



# Temporal dynamics of tree xylem water isotopes: In-situ monitoring and modelling

Stefan Seeger and Markus Weiler

Hydrology, Faculty of Environment and Natural Resources, University of Freiburg

**Correspondence:** Stefan Seeger (stefan.seeger@hydrology.uni-freiburg.de)

**Abstract.** We developed a setup for a fully automated, high frequency in-situ monitoring system of the stable water isotopes Deuterium and $^{18}$O in soil water and tree xylem. The setup was tested for 12 weeks within an isotopic labelling experiment during a large artificial sprinkling experiment including three mature European beech (*Fagus sylvatica*) trees. Our setup allowed for one measurement every 12–20 minutes, enabling us to obtain about seven measurements per day for each of our 15 in-situ
probes in the soil and tree xylem. While the labelling induced an abrupt step pulse in the soil water isotopic signature, it took seven to ten days until the isotopic signatures at the trees' stem bases reached their peak label concentrations and it took about 14 days until the isotopic signatures at 8 m stem height levelled off around the same values. During the experiment, we observed the effects of several rain events and dry periods on the xylem water isotopic signatures, which fluctuated between the measured isotopic signatures observed in the upper and lower soil horizons. In order to explain our observations, we combined
an already existing root water uptake (RWU) model with a newly developed approach to simulate the propagation of isotopic signatures from the root tips to the stem base and further up along the stem. The key to a proper simulation of the observed short term dynamics of xylem water isotopes, was accounting for sap flow velocities and the flow path length distribution within the root and stem xylem. Our modelling framework allowed us to identify parameter values that relate to root depth, horizontal root distribution and wilting point. The insights gained from this study can help to improve the representation of stable water
isotopes in trees within ecohydrological models and the prediction of transit time distribution and water age of transpiration fluxes.

## 1 Introduction

Transpiration from terrestrial plants is a key component of the global hydrological cycle and its fraction of the total water balance might even increase under projected future climatic conditions (Bernacchi and VanLoocke, 2015). Process based
ecohydrological models can be an important tool to gain realistic estimates of the vegetation's response to climatic changes. Such process based models need detailed data on plant water uptake and transpiration. These processes can and have been studied intensively with the help of stable water isotopes (White et al., 1985; Calder, 1991; Dawson and Ehleringer, 1991; Busch et al., 1992; Ehleringer and Dawson, 1992; Zhang et al., 1999; Dawson et al., 2002).



## 1.1 Isotope tracer enhanced observations of tree water dynamics

To observe the temporal dynamics of water within the soil-plant-atmosphere continuum (SPAC), isotopic labelling experiments offer a unique opportunity to create distinct pulses that can be traced from the soil, through the plant, to the atmosphere. For small experimental plots, chamber based measurements can capture the isotopic composition of transpiration ($\delta_T$) following isotopically labelled irrigation pulses (e.g. Yepez et al. (2005); Volkmann et al. (2016a)). Due to dimensional limits set by the size of the required chamber, this method is not applicable to adult trees.

Larger scale irrigation experiments were conducted in green house experiments with around 15 m high tropical trees (Evaristo et al., 2019) and in a Central European forest with grown, 25 m high, specimen of *Fagus sylvatica* and *Abies alba* (Magh et al., 2020). Instead of chamber measurements of $\delta_T$, they extracted water from tree crown branches to monitor the isotopic composition of xylem water ($\delta_{xyl}$). For seven months, Evaristo et al. (2019) had a weekly $\delta_{xyl}$ sampling scheme, while Magh et al. (2020) started with a sub-daily $\delta_{xyl}$ sampling scheme for five days and continued the following two month with a
sampling frequency of four to six days. Both experiments observed notable delays (days to weeks) between tracer application and detection within the sampled crown branches.

A more specific investigation of tree water dynamics can be achieved by skipping soil and roots and directly injecting small amounts of $D_2O$ into the stem base (James et al., 2003; Meinzer et al., 2006; Schwendenmann et al., 2010; Gaines et al., 2016). Due to the high Deuterium concentrations of the injected label, tracer breakthrough curves within the tree crown
could be acquired by analysing condensate from branch or foilar samples placed into zipper bags. Time delays between tracer application and maximum tracer concentrations within the tree crowns mostly have been reported to amount to a few days, but for a 50 m high specimen of *Tsuga heterophylla* Meinzer et al. (2006) also reported a delay of about 30 days. A comparison of heat tracing derived sap flux velocities with isotope tracing derived velocities revealed that the latter may be 4 to 16 times higher (Meinzer et al., 2006; Gaines et al., 2016).

## 45 1.2 Water uptake and tree isotope modelling

Rothfuss and Javaux (2017) have reviewed 159 studies combining root water uptake (RWU) and isotopes. 46% of those studies used isotopic data to infer a specific soil layer or water pool as source of RWU. Another 50% of those studies were using two- or multi-source linear mixing models of varying statistical sophistication. Only the remaining 4% of the reviewed studies were using physically based analytical or numerical models. This means, that the vast majority of isotope based RWU studies is
based on graphic or statistical methods, which allow a description of observations but abstain from offering physically-based models, which could be used to predict the reaction of RWU to changing environmental conditions.

Lv (2014), Rabbel et al. (2018) and Brinkmann et al. (2018) have modelled RWU of trees with implementations of a Feddes-style RWU model described by Feddes et al. (1976) and Jarvis (1989) as implemented within the soil hydrological model HYDRUS-1D (Šimůnek et al., 2008). Even though the Feddes RWU model was originally developed to model water
uptake of agricultural crops, the above listed applications demonstrated that Feddes RWU models are also suited to simulate RWU of mature trees. Brinkmann et al. (2018) used a Feddes RWU model, driven with soil water potentials and soil $^{18}O$





concentrations simulated with an isotope enabled version of HYDRUS-1D (Stumpp et al., 2012), to compute $\delta^{18}$O signatures of RWU. These predictions compared well to fortnightly sampled $\delta^{18}$O values of branch xylem water.

The assumption that isotopic signatures of RWU ($\delta_{RWU}$) are directly related to simultaneously sampled $\delta_{xyl}$ has been chal-

lenged by studies of Knighton et al. (2020) and de Deurwaerder et al. (2020). Knighton et al. (2020) juxtaposed a "zero storage case" ($\delta_{RWU} = \delta_{xyl}$) and two alternative cases (well mixed & piston flow) that included tree internal water storage. Model results for the cases with tree internal water storage compared better to observational data than the zero storage case. Due to a limited temporal resolution of their observed data, Knighton et al. (2020) were not able to conclude whether the piston flow or the well mixed case were more appropriate and hypothesized that the actual behaviour of the system may lie between those two

cases. de Deurwaerder et al. (2020) proposed a model that comprised a soil water potential driven RWU component and a stem water transport module that is based on an advection-diffusion model coupled to sap flow velocities. Based on their model, de Deurwaerder et al. (2020) predicted that diel fluctuations of $\delta_{RWU}$ (caused by fluctuations in leaf water potential) should be transmitted from the stem base upwards, effectively causing periodic patterns of $\delta_{xyl}$ along the stem height.

### 1.3 Measurement of soil and xylem water isotopes

Until recently, measurements of $\delta_{xyl}$ and isotopic compositions of soil water ($\delta_{soil}$) usually required the laborious extraction of the respective waters. Scholander pressure chambers (Scholander et al., 1965) can be used to extract xylem water from branches (Rennenberg et al., 1996; Magh et al., 2020). Another common xylem water extraction method is cryogenic vacuum extraction Dawson and Ehleringer (1991); Orlowski et al. (2013); Newberry et al. (2017), which is also commonly used for soil pore water extraction Dalton (1988). A detailed review on available soil pore water extraction techniques was done by Orlowski

et al. (2016). All of these methods require destructive sampling of plant or soil material and subsequently careful sample storage and treatment. The required effort per sample and the disturbance caused by each sampling limits the total number of samples as well as the maximum sampling frequency. In fact, the very nature of destructive sampling renders repeated measurements from the exact same spot impossible. Consequently, a time series generated with a destructive sampling method will inevitably also be affected by spatial variability.

With the advent of laser spectroscopy, the measurement of stable water isotopes does not any longer require the extraction of liquid water samples. Instead, the isotopic composition of the vapour contained in a gaseous sample can directly be analysed in the lab and even in the field. Firstly, this leads to the development of equilibration based lab methods which allow for indirect water isotope measurements from samples without the need for the extraction of liquid water from destructive samples (Wassenaar et al., 2008; Garvelmann et al., 2012). Subsequently, different approaches for in-situ sampling of stable water

isotopes have been developed. For a thorough review of in-situ water isotope sampling techniques we would like to refer to Beyer et al. (2020).

Different approaches for in-situ measurements of $\delta_{soil}$ have been proposed and tested by Soderberg et al. (2012), Rothfuss et al. (2013), Volkmann and Weiler (2014) and Gaj et al. (2016). So far the method of Rothfuss et al. (2013) has seen the most subsequent applications ranging from long term, continuous lab-experiments (Rothfuss et al., 2015) to campaign based

field studies (Oerter and Bowen, 2017; Kübert et al., 2020). For in-situ sampling of $\delta_{xyl}$, Marshall et al. (2020) developed the



bore hole equilibration method, which goes entirely without a specific probe and instead connects tubing directly to a borehole through a stem. So far, this method only has been tested on tree logs and in a greenhouse experiment with small trees placed in pots filled with water instead of soil.

The in-situ probes developed by Volkmann and Weiler (2014) are the only approach that has been proven to work in soil (in six two-day sampling periods, Volkmann et al. (2016a)) as well as within tree xylem (in two young Maple trees over 11 days, Volkmann et al. (2016b)). While these probes have been called SWIPs (soil water isotope probes) when used in the soil and XWIPs (xylem water isotope probes) when used in tree xylem, the actual probes in our study are identical in both use cases. Therefore we propose an alternative third name: WIP (water isotope probe), which should encompass both of the mentioned and all further use cases of this particular probe design.

This study's primary motivation was to test out the automated measurement setup in a prolonged field experiment. Nevertheless, the results of our WIP measurements and the subsequent analyses could also be used to scrutinize the following two hypotheses regarding tree water isotope dynamics and transit times:

1. Xylem water isotopic signatures are equivalent to the isotopic signature of root water uptake.

2. The spatial distribution of a tree's root system has an effect on the shape of the xylem water age (i.e. time elapsed since uptake) distribution, which in turn can be used to infer information on that spatial distribution of the root system.

## 2 Methods

### 2.1 Theoretical basis

#### 2.1.1 Root water uptake model

The isotopic concentration of root water uptake ($\delta_{\text{RWU}}$) for either $^{18}$O or Deuterium can be computed with the following equation:

$$\delta_{\text{RWU}} = \frac{\sum_{i=1}^{N} \delta_{\text{i}} \overline{S_i} \Delta z_i}{\sum_{i=1}^{N} \overline{S_i} \Delta z_i} \tag{1}$$

where $i$ is the index of a specific soil layer, $N$ is the total number of all soil layers, $\delta_{\text{i}}$ is the soil water isotope signature and $\Delta z_i$ the thickness of soil layer $i$. According to Jarvis (1989), $\overline{S_i}$ is the relative sink strength of the soil layer $i$ which is defined by:

$$\overline{S_i} = R_i \alpha_i \tag{2}$$

where $R_i$ is the proportion of total fine root length within layer $i$ and $\alpha_i$ is a stress index. Following a root distribution model introduced by Gerwitz and Page (1974), Jarvis (1989) defined $R_i$ as:

$$R_i = f \left( \frac{\Delta z_i}{z_{\text{r}}} \right) exp \left( -f \frac{z_i}{z_{\text{r}}} \right) \tag{3}$$





In this equation the tuning parameters $z_\mathrm{r}$ and $f$ are not independent from each other - different combinations of the two
parameters lead to identical root distributions. For all further considerations, we fixed the value of $f$ to 3, which causes $z_\mathrm{r}$ to be
the depth above which 95% of all roots are located.

Jarvis (1989) computes the stress index $\alpha$ is as:

$$
\alpha_i = \begin{cases} \frac{1-\overline{\theta}_i}{1-\overline{\theta}_{c2}}, & \overline{\theta}_{c2} < \overline{\theta}_i \leq 1 \\ 1, & \overline{\theta}_{c1} \leq \overline{\theta}_i \leq \overline{\theta}_{c2} \\ \frac{\overline{\theta}_i}{\overline{\theta}_{c1}}, & 0 \leq \overline{\theta}_i \leq \overline{\theta}_{c1} \end{cases} \tag{4}
$$

where $\overline{\theta}_{c1}$ and $\overline{\theta}_{c2}$ are critical values that define a trapezoidal function that relates the normalized volumetric soil water content
$\overline{\theta}$ to the water stress index $\alpha$. $\overline{\theta}$ is computed according to:

$$
\overline{\theta}_i = \frac{\theta_i - \theta_w}{\theta_s - \theta_w} \tag{5}
$$

where $\theta_i$ is a soil layers volumetric water content, that lies between wilting point $\theta_w$ and saturation $\theta_s$.

### 2.1.2 Flow path length distribution

Root water uptake (RWU) happens at different depths and different radial distances from the stem base. An isotopic signature
measured at the stem base will consequently represent a mixture of waters transported over various distances (or flow path
lengths) from the root tips to the stem. The flow path length distribution (FPLD) of a tree root system is determined by (1) a
vertical component $f(z)$, depending on the soil depth $z$ and (2) a radial component $g(r)$, depending on the radial distance from
the stem center $r$.

The depth dependent RWU probability function $f(z)$ can be defined by a mathematical function (like in Eq. 3) or by
empirical data of vertical fine root density distributions. The radial RWU density function $g(r)$ is, however, much less reported
and studied. In most cases, RWU is considered from a one dimensional perspective with regard to depth alone. For our purposes
we are also interested in the distribution of fine roots regarding the radial distance from the stem. Due to a lack of reported
observational data, we propose the following equation to describe the relative root density (integrated over all depths) along a
radial transect between the centre of the stem ($r = 0$) and the maximum radial extent of the rooting system ($r = r_\mathrm{max}$):

$$
g_r(r) = \frac{e^{1/\lambda} - (e^{r/r_\mathrm{max}})^{1/\lambda}}{e^{1/\lambda} - 1} \tag{6}
$$

where the radial density of roots decreases linearly towards the outer extent of the rooting system when distance decay parameter $\lambda = 1$ and slower for $\lambda < 1$. In order to account for the effect of the projected area of a certain distance class, $g(r)$ is
computed according to:

$$
g(r) = \frac{g_r(r)\frac{r}{r_\mathrm{max}}}{\int_0^{r_\mathrm{max}} g_r(x)\frac{x}{r_\mathrm{max}}\mathrm{d}x} \tag{7}
$$

where the denominator term normalizes the integral of $g(r)$ within $0 \leq r \leq r_\mathrm{max}$ to unity.





With the vertical component $f(z)$ and the radial component $g(r)$ defined, both can be combined into a probability density function of RWU $h_{\mathrm{R}}(z,r)$ in the following way:

$$h_{\mathrm{R}}(z,r) = \begin{pmatrix} f(z_1)g(r_1) & f(z_1)g(r_2) & \cdots & f(z_1)g(r_{j-1}) & f(z_1)g(r_j) \\ f(z_2)g(r_1) & f(z_2)g(r_2) & \cdots & f(z_2)g(r_{j-1}) & f(z_2)g(r_j) \\ \cdots & \cdots & \cdots & \cdots & \cdots \\ f(z_{i-1})g(r_1) & f(z_{i-1})g(r_2) & \cdots & f(z_{i-1})g(r_{j-1}) & f(z_{i-1})g(r_j) \\ f(z_i)g(r_1) & f(z_i)g(r_2) & \cdots & f(z_i)g(r_{j-1}) & f(z_i)g(r_j) \end{pmatrix} \tag{8}$$

where $z_1, z_2, \ldots z_{i-1}, z_i$ and $r_1, r_2, \ldots r_{j-1}, r_j$ denote sufficiently fine spaced values of $z$ and $r$ within the respective domains of

$f(z)$ and $g(r)$. Now we can use the Pythagorean theorem to compute the total distance to the stem base and aggregate $h_{\mathrm{R}}(z,r)$ to the flow path length distribution $h_{\mathrm{R}}(s)$:

$$h_{\mathrm{R}}(s) = \sum h_{\mathrm{R}}(z,r), \text{for all } z \text{ and } r \text{ that fulfill} \sqrt{z^2 + r^2} = s \tag{9}$$

### 2.1.3 Signal transformation and convolution

Inspired by a long line of tracer hydrological research (e.g.: Małoszewski and Zuber (1982); Kirchner et al. (2000); Weiler et al.

(2003); McGuire and McDonnell (2006)) that uses convolution integrals to transfer precipitation tracer time series to stream tracer time series via a transfer function, we adapt this approach to model the tracer dynamics within the tree xylem.

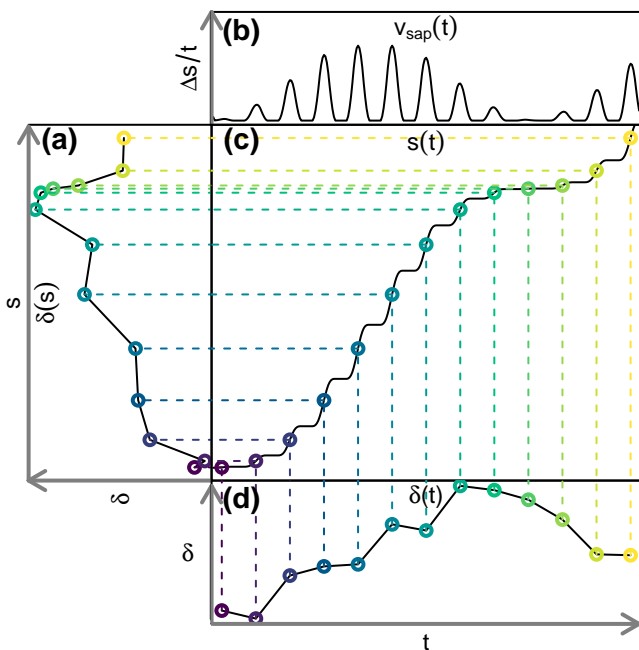

**Figure 1.** Exemplary transformation of **(d)** a tracer concentration time series $\delta(t)$ to **(a)** a concentration function depending on the cumulative sap flow distance $\delta(s)$. A time series of **(b)** sap flow velocities $v_{\mathrm{sap}}(t)$ is needed for the construction of **(c)** the auxiliary function $s(t)$.





In order to neutralize the effects of sap flow velocity variations we transform our observed tracer time series $\delta(t)$ to "sap distance series" $\delta(s)$. This transformation is depicted in Fig. 1 and it requires to obtain a cumulative sap flow distance $s$ for each time step p $t$ with the following equation:

$$s = \sum_{i=1}^{n} \Delta s_i = \sum_{i=1}^{n} \Delta t_i (\overline{\Delta s_i / \Delta t_i}) = \sum_{i=1}^{n} \Delta t_i \overline{v_i} \qquad (10)$$

where $\Delta s_i$ is the distance travelled by the sap during $t_i$, and $\overline{\Delta v_i}$ is the mean sap flow velocity of $t_i$, which has a duration of $\Delta t_i$. With an established transformation between time $t$ and sap flow distance $s$, we can transform observed tracer time series $\delta(t)$ to tracer "sap flow distance" series $\delta(s)$.

After the transformation of $\delta(t)$ to $\delta(s)$, we can predict the isotopic signature at the stem base, $\delta_{\mathrm{xyl.R}}$, by convolving $\delta_{\mathrm{RWU}}$ with the FPLD between the stem base and all root tips (e.g. $h_{\mathrm{R}}$ from Sec. 2.1.2):

$$\delta_{\mathrm{xyl.R}}(s) = \int_{0}^{\infty} h_{\mathrm{R}}(s')\delta_{\mathrm{RWU}}(s - s')\mathrm{d}s' \qquad (11)$$

Similarly, we can relate $\delta_{\mathrm{xyl.R}}$ to $\delta_{\mathrm{xyl.H}}$ (an isotopic stem xylem signature at a certain height above the stem base) by convolution with an appropriate transfer function that manages to represent the respective FPLD.

Eventually the methodology described in the previous sections can be combined as follows:

– transform input tracer time series to input tracer sap distance series

– convolve input tracer sap distance series with an appropriate transfer function (representing a static FPLD)

– transform output tracer sap distance series back into an output tracer time series

## 2.2  Field experiment

### 2.2.1  Study site and instrumentation

The experiment described in this study took place at a research site located on a 25 degree steep hillslope of the Swabian Alb in Southwestern Germany (47.98° N, 8.75°E , close to the city of Tuttlingen). The soil type is a Rendzic Leptosol developed on glacial slope debris of Jurassic limestone. Soil texture in the upper 70 cm ranges from silty clay to silty loam with a fraction of up to 43% of rocks. Larger rock fragments can be found as shallow as 20 cm below the soil surface and become very abundant below 50 cm. Our 200 m$^2$ experimental plot was situated within a stand of 90–100 year old European beech (Fagus sylvatica) with diameters at breast height ranging from 45 to 90 cm and contained a total of four beech trees, of which three have been instrumented.

A sketch of the experimental setup is shown in Fig. 2, which omits the repeated instances of the principal components and sensors. Precipitation (throughfall) samples were taken as often as possible from four rain gauges (Fig. 2.A) distributed on and around the study plot. One of those gauges was combined with a tipping bucket (0.2 mm resolution, Davis Instruments, Hayward, USA) in order to register hourly precipitation amounts. A total of four depth profiles of volumetric soil moisture





sensors (SMT100, Truebner GmbH, Neustadt, Germany) were installed in depths of 10, 20, 40 and 60 cm (Fig. 2.G) with different radial distances to the trees.

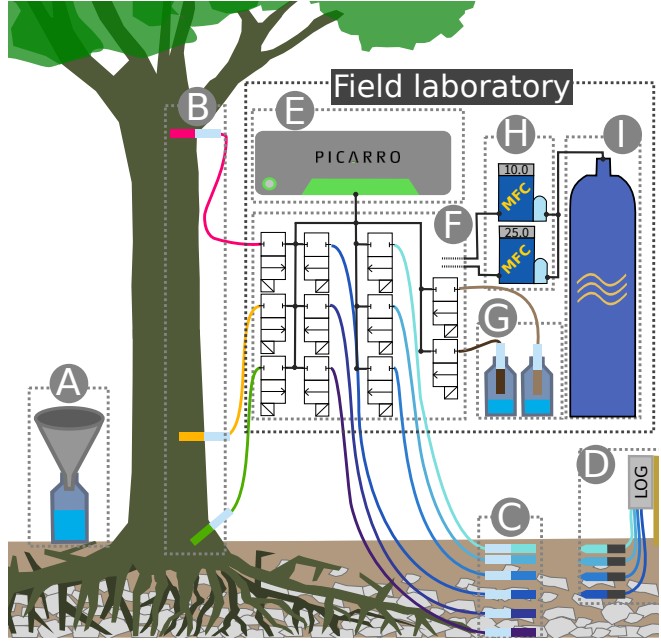

**Figure 2.** Sketch of the experimental setup: A) Throughfall sampler; B) WIPs at 10, 150 and 800 cm stem height; C) WIPs in 10, 20, 40, 60, 80 and 100 cm soil depth; D) Volumetric soil moisture sensors in 10, 20, 40 and 60 cm depth; E) CRDS for stable water isotope measurements; F) Valve system to sequentially connect each probe to the CRDS and the two mass flow controllers; G) Standard probes in headspace of liquid water standards; H) Mass flow controllers; I) Dry air for sample dilution and through-flow

At depths of 10, 20, 40, 60, 80 and 100 cm we installed one profile of WIP into the soil (Fig. 2.F). Additionally we installed WIPs into the xylem of two Beech trees (T1 and T2) at 10, 150 and 800 cm height (Fig. 2.B), as well as into the xylem of another Beech tree (T3) at 150 cm height. The different installation heights are reflected by the probe IDs (e.g. T1R, T1B and T1H), where R stands for "root" (10 cm), B for "breast height" (150 cm) and H for "high" (800 cm). Depending on the locations of the probes and the trees, the tubing lengths between the probes and the CRDS varied between 5 m (T2R and T2B) and 20 m (T1H). Additionally, we put two probes into the head space of two sealed 1-Liter containers made of high-density polyethylene (HDPE), filled with 250 mL of water with known isotopic composition (Fig. 2.E). These two probes acted as our light ($\delta 18O$ = -11.61‰, $\delta D$ = -82.4 ‰) and heavy ($\delta 18O$ = -0.93 ‰, $\delta D$ = 6.63‰) reference standards and were placed directly next to the CRDS in our field lab.

In order to install the WIPs with a diameter of 10 mm into the tree xylem, we drilled a hole with a 10 mm wood drill, a strip of tape marking a hole depth of slightly more than the length of the 5 cm porous head. Care was taken, not to overheat the drill in order to avoid singed xylem wood in the hole. Afterwards, a 10.2 mm metal drill was used to slightly widen the hole and clear out wood chip residue. Then the probe was firmly pushed into the hole just deep enough to place the porous head behind





the the phloem. In a final step, silicone was applied around the probe to seal the hole. While the silicone is curing, organic fumes may interfere with the measurements of the CRDS. Therefore it is advisable, to install the probes several days before the scheduled start of the measurements. In addition, we placed heat pulse based sap flow sensors (East 30 Sensors, Pullman USA) in the vicinity of each WIP. Measurements of the tipping bucket, soil moisture sensors and sap flow sensors where logged in
10 minute intervals with a CR1000 data logger (Campbell Scientific, USA).

### 2.2.2 Tracer Experiment

The stable water isotope concentrations for $^{18}$O and Deuterium in this paper are noted in the $\delta$-notation relative to Vienna standard mean ocean water (VSMOW):

$$\delta_{\text{sample}} = \left( \frac{R_{\text{sample}}}{R_{\text{VSMOW}}} - 1 \right) \times 1000\text{‰} \tag{12}$$

where $R_{\text{sample}}$ and $R_{\text{VSMOW}}$ are the D/H or $^{18}$O/$^{16}$O ratios of the sample and VSMOW, respectively.

Since not direct water source was available to irrigate the plot with a defined amount of 150 mm, 60 m$^3$ groundwater were trucked to the site, run through an industrial deionizer (VE-300 (6x50 Liter), AFT GmbH & Co.KG) to reduce the mineral content to low levels typically found in natural rainfall. In our case the sprinkling water had an electrical conductivity of around 20 μS/cm and a isotopic composition of ($\delta^{18}$O = -9‰, $\delta$D = -63‰). By mixing 1 kg of D$_2$O with the 60,000 Liters
of water within a collapsible pillow tank (custom made by Faltsilo GmbH, Bad Bramstedt, Germany), which was placed 100 m upslope of the experimental plot, we obtained Deuterium enriched irrigation water ($\delta^{18}$O = -9‰, $\delta$D = 40‰). On 21 May 2019, we used an array of 6 sprinklers (XCEL-Wobbler by Senninger, Clermont, USA) driven by the height difference between pillow tank and irrigation site, to distribute our prepared D$_2$O-groundwater-mixture onto the experimental plot. The amount of irrigation water actually reaching the core plot area of 10 by 20 m was equivalent to 150 mm rainfall within 8 hours. After this
artificial event, we continued to monitor the plot under natural conditions for another 12 weeks.

### 2.2.3 Stable water isotope measurement system

On two days prior to the irrigation, as well as one day after the irrigation, we took a total of five destructive soil core samples. We used an electric breaker (HM1812, Makita Werkzeug GmbH, Germany) to drive a core probe (60 x 1000 mm, Geotechnik Dunkel GmbH & Co. KG, Hergolding, Germany) into the soil until we hit larger rocks. The soil cores were extracted and
split into 10 cm segments, yielding 120 to 300 g of fine soil and skeleton material per depth increment, that were filled into aluminum coated coffee bags (WEBAbag CB400-420siZ, Weber Packaging GmbH, Güglingen, Germany).

Following the equilibration bag method after Wassenaar et al. (2008) and Garvelmann et al. (2012), the sample bags were filled with dehumidified air in the lab and permanently sealed with sealing tongs (Weber Packaging GmbH). After 24 hours of equilibrium at constant temperature, the sample bags were punctured by a hollow needle connected to the inlet port of a
Cavity Ring Down Spectrometer (CRDS) stable water isotope analyser (L2120-I, Picarro, Santa Clara, USA). After five to ten minutes, the isotope analyser readings were reaching plateaus of constant values. Before and after the measurements of the



soil bags, we also measured three standard bags filled with liquid water of known isotopic composition, which were treated identical to the bags containing soil samples.

The WIPs used in this study were build at the Chair of Hydrology of the University of Freiburg, Germany, following the
"diffusion-dilution sampling" (DDS) design described by Volkmann and Weiler (2014). Figure 3 shows a sketch of a WIP installed into the soil. Key element of a WIP is the mixing chamber (custom product manufactured by Horst Fischer GmbH, Gundelfingen, Germany) right behind the porous membrane head (custom product manufactured by Porex Technologies, Aachen, Germany), where a dilution line dilutes the sample gas in order to prevent condensation within the sampling line on its way to the isotope analyser. The additional through flow line compensates for any pressure differences arising from dilution rates that
are smaller than the isotope analyser's sample rate and to allow for flushing the system with dried air.

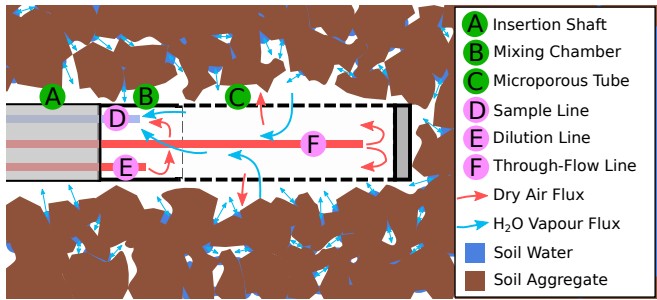

**Figure 3.** Sketch of a WIP installed into soil. Water vapour from the surrounding medium (in this case soil) diffuses through the microporous membrane (C). Before being sucked into the sample line (D), the sample gas is diluted within the mixing chamber (B).

Deviating from the original applications of WIPs (Volkmann and Weiler (2014), Volkmann et al. (2016a) and Volkmann et al. (2016b)), we switched from using nitrogen as dilution and through-flow gas to compressed dry air. As Gralher et al. (2018) have shown, intrusion of ambient air has a considerably larger influence on measurements relying on N2, compared to measurements that use dry air as dilution and flush medium. Furthermore, transporting compressed dry air in a vehicle underlies
less restrictions than transporting compressed N2. A pressure regulator reduced the pressure at the outlet of the compressed dry air bottle down to 1.5 bar. From there on the dry air stream was split amongst two mass flow controllers (GFC17, 0- 50mL/min and MFC 35828, 0-200 mL/min, both supplied by AnalytMTC GmbH, Müllheim, Germany) (Fig. 2).

Over custom manufactured valve manifolds (Horst Fischer GmbH, Gundelfingen, Germany), equipped with 2-way electric valves (EC-2M-12, Clippard, Cinncinati, USA), each of the probes was connected to the two mass flow controllers and the
sample inlet of the field deployed CRDS stable water isotope analyser (L1102-i, Picarro, Santa Clara, USA). Connections between the probes, valve manifolds, mass flow controllers and isotope analyser were made with 1/16" FEP (inner diameter = 0.75 mm) tubing (Techlab GmbH, Braunschweig, Germany) and Flangeless Fittings (XP-220, IDEX, Lake Forest, USA). Connections between the dry air supply and the mass flow controllers were made with stainless steel fittings (Swagelok, Solon, USA). We set up our field deployed CRDS and its peripherals within a watertight container and supplied it with electricity
from a nearby power line.





Based on the Arduino microcontroller platform, we designed and built a custom circuit board that is able to switch electro-magnetic valves and to provide two independent analogue voltages. Those voltage signals were used to control the throughflow of two mass flow controllers. A custom made Python based software GUI that is able to interface the Arduino based circuit board and interpret the CRDS' log files near real time enabled us to automate the measurement process to a large extent and to
quickly adapt flow rates and times for flushing and measuring as well as the order of the probe sequence. We attached a USB-modem (E531, Huawei Technologies, Shenzen, China) to the isotope analyser to regularly transmit summarized measurement results (less than 20 kB/h) to an ftp-server. With this setup, we could remotely monitor, but not interfere with the ongoing measurements. Further details on this automation system (circuit board designs, assembly instructions and source codes of the control software) can be found in in the following online repository: https://github.com/stseeger/IsWISaS.

In order to obtain a measurement value for a certain probe, we activated the three respective valves of that probe (sample, dilution and through-flow line). The time of activation was saved to an automatically generated log file. At the same time, we initiated a flush phase by setting the dilution rate to the same as the CRDS' sample intake rate (in our case 35 mL/min) and setting the flow rate in the through-flow line to zero. The duration of the flush phase was chosen depending on the overall tubing length of the probe – from three minutes for probes with short tubing, up to ten minutes for probes with 20 m long tubing.
After the flush phase, we started the measurement phase by reducing the dilution flow rate to 10 mL/min and increasing the through-flow rate to 25 mL/min – dilution and through-flow yet again adding up to the constant sample flow rate of the CRDS. The measurement phase either ended after a fixed amount of time (in our case 20 minutes) or earlier, in case the measured raw values of the CRDS approached a stable plateau. Plateau detection was automated by checking standard deviations $s$ and a trend index $I_T$ of the last two minutes of CRDS raw data for $H_2O$ (sample moisture content), $\delta^{18}O$ and $\delta D$ against the
thresholds listed in Table 1. The standard deviation $s$ was computed as:

$$s = \sqrt{\frac{1}{N-1}\sum_{i=1}^{N}(x_i - \overline{x})^2} \tag{13}$$

where N is the number of raw instrument readings within th last two minutes, $x_1, x_2, ..., x_N$ are all of the respective values and $\overline{x}$ the mean of all these values. With the value of $N/2$ rounded to a whole number, the trend index $I_T$ was computed as:

$$I_T = \left| \frac{\sum_{i=1}^{N/2} x_i}{N/2} - \frac{\sum_{i=N/2+1}^{N} x_i}{N - N/2} \right| \tag{14}$$

**Table 1.** Threshold values for standard deviation $s$ and trend index $I_T$ used for automated detection of stable measurement values.

| Parameter (unit) | Threshold values | |
| --- | --- | --- |
| | $s$ | $I_T$ |
| $H_2O$ (ppmV) | 200 | 150 |
| $\delta^{18}O$ (‰) | 0.2 | 0.1 |
| $\delta D$ (‰) | 1 | 0.5 |





After each probe measurement, the system automatically proceeded to flush and measure the next probe within the specified probe sequence, which was automatically restarted upon its completion.

## 2.3   Data processing

From the raw sensor data, we computed the sap flow velocity $v_{\text{sap}}$ in $\text{m/s}$ according to Campbell et al. (1991):

$$v_{\text{sap}} = \frac{2k}{C_w(r_u + r_d)} \, ln\left(\frac{\Delta_{Tu}}{\Delta_{Td}}\right) \tag{15}$$

where $k$ is the thermal conductivity of sapwood set to $0.5 \, \text{W m}^{-1} \, \text{K}^{-1}$ (Hassler et al., 2018), $C_w$ is the specific heat capacity of water at $4.184 \times 10^6 \, \text{m}^3 \, \text{K}^{-1}$, $r_u$ and $r_d$ (both = 6mm) are the distances of the central heater needle to the up- and downstream thermistor needles and $\Delta_{Tu}$ and $\Delta_{Td}$ refer to the temperature changes induced by the heating pulse at the up- and downstream needles, respectively.

Our measured sap flow time series was limited from May 2019 to 8 August 2019. In order to extend it to the whole year,
we assumed no sap flow during the time where our deciduous trees did not have any foliage (before May and after October). During those periods, we set $v_{\text{sap}}$ to $0 \, \text{m/s}$. For the remaining period between 8 August and 31 October, we fitted the two parameters $a$ and $b$ of the following equation:

$$v_{\text{sap}} = \text{VPD} \times a/(\text{VPD} + b) \tag{16}$$

where VPD is the vapor pressure deficit (derived from meteorological data of the nearby meteorological site *Klippeneck* of
the German Weather service *DWD*). In order to account for decreasing $v_{\text{sap}}$ due to leaf senescence, we multiplied the estimates obtained by Eq. 2.3 with a linearly decreasing reduction factor between mid of September and end of October. Urban et al. (2015) reported a comparable autumnal reduction of sap flow in relation to potential evaporation for Central European beech trees.

As an additional mean to investigate RWU independently of isotope measurements, we followed Guderle and Hildebrandt
(2015) and used measurements of volumetric soil moisture – more specifically the daily decline of soil moisture during dry days – as an indicator of RWU. To compare the soil moisture measurements with the RWU model, we derived a water uptake ratio $r_{\text{RWU}}$ in two ways:

$$r_{\text{RWU}} = \frac{\Delta_{\theta u}}{\Delta_{\theta l}} = \frac{\overline{S}_u}{\overline{S}_l} \tag{17}$$

where the first ratio defines the measured daily decline of soil moisture at the upper soil layer $\Delta_{\theta u}$ (average of 10 and 20
cm depth) and $\Delta_{\theta l}$ the equivalent for the lower soil layer (average 40 and 60 cm depth). The second ratio defines the relative modelled uptake strengths of Eq.2 (with $\overline{S}_u$ as the average of $\overline{S}_i$ for the depths of 10 and 20 cm and $\overline{S}_l$ as the average of $\overline{S}_i$ for the depths of 40 and 60 cm). The soil moisture based computation of $r_{\text{RWU}}$ is limited to days where the soil water content throughout the depth profile is below field capacity.

In order to analyse the WIP measurements, we used the recorded valve switching times to aggregate the raw CRDS log file
data by computing average values for the last two minutes of each period. The parameters of interest were sample water vapour





content, $\delta^{18}$O and $\delta$D. In the next step we corrected for the influence of temperature on the fractionation factors during vapour equilibration at the probe head. Instead of direct temperature measurements, we relied on the assumption that the temperature at the place of equilibration (i.e. around the probe head) is reflected by the water vapour content of the obtained sample gas – given that the sample rate and the dilution rate are held constant. By computing linear regressions between measured vapour

isotope values ($\delta_m$) for our two standards and the sample gas moisture contents ($C_m$), we derived the slopes needed to correct all vapour isotope measurements to one reference moisture content value ($C_r = 18000$ ppmV) according to the following equation:

$$\delta_v = \delta_m - \Delta_{C\delta}(C_r - C_m) \tag{18}$$

where $\delta_v$ is the corrected isotope value and $\Delta_{C\delta}$ is the slope obtained by the linear regression between $C_m$ and $\delta_m$ values of

the standards.

To infer the isotopic signature of the liquid water that equilibriated with the sampled vapour, we used the relationship between the known liquid phase values of our two standards and the respective observed vapor values:

$$\delta_{l.x} = \frac{\delta_{v.x} - \delta_{v.L}}{\Delta_{LH}} + \delta_{l.L} \tag{19}$$

where $\delta_{l.x}$ is the normalized liquid phase isotopic value of measurement $x$, $\delta_{v.x}$ is the moisture corrected vapour value of

measurement $x$, $\delta_{v.L}$ is the moisture corrected vapour value of the light standard, $\delta_{l.L}$, is the liquid phase isotopic value of the light standard and $\Delta_{LH}$ is the slope obtained by:

$$\Delta_{LH} = \frac{\delta_{v.L} - \delta_{v.H}}{\delta_{l.L} - \delta_{l.H}} \tag{20}$$

with $\delta_{v.H}$ as the moisture corrected vapour value of the heavy standard and finally $\delta_{l.L}$ and $\delta_{l.H}$ as the known liquid water isotope values of the light and heavy standards, respectively.

Under stable environmental conditions, $\delta_{v.L}$ and $\delta_{v.H}$ should not change at all, but in a field experiment more frequent measurements of these standards are highly recommended. We treated both standards as regular parts of our measurement sequence, yielding one measurement of each standard every three to four hours. For the normalization procedure we interpolated between those actually measured standard values in order to estimate the standard values for each measurement.

### 2.3.1   Modelling RWU, FPLDs and xylem water age

To obtain the necessary input data for RWU modelling, we interpolated our observations of volumetric soil moisture and soil isotopic signatures over time and space to generate a continuous time series in time and space. Lacking any observations below 60 cm for volumetric soil moisture and 1 m for soil water isotopes, we assumed constant boundary conditions ($\theta = 25$ %, $\delta^{18}$O = -10.5 ‰ and $\delta$D = -73 ‰) at the lower profile border in 2 m depth.

Subsequently, we used the Jarvis RWU model (see 2.1.1) to compute RWU amounts and isotopic signatures and subsequently

the convolution approach described in (2.1.2) in order to transfer the RWU isotopic signatures to the respective stem base xylem isotopic signatures, which could be compared to our in-situ xylem water measurements at the stem base. Model performance





was evaluated by computing the the root mean square error RMSE between model predictions and observations for Deuterium and $^{18}$O at the stem base as well as for the water uptake index $r_U$ (see 2.3). With $\hat{y}_i$ as the i$^{\text{th}}$ of a total of $n$ observations and $y_i$ as the respective simulated value, the RMSE was computed according to:

$$\text{RMSE} = \sqrt{\frac{\sum_{i=1}^{n}(\hat{y}_i - y_i)^2}{n}} \qquad (21)$$

Next, we evaluated the model for 500 random parameter sets within the value ranges given in Table 2. The soil related model parameters were assumed to be identical over the whole profile depth. Based on the observed soil moisture time series we set the volumetric soil moisture at saturation $\theta_s$ to a fix value of 45%. Since soil moisture levels near saturation only occurred for a short time during the irrigation, we set the model parameter $\overline{\theta}_{c2}$ to a value of 100%.

Eventually, to relate modelled RWU to our observed xylem isotopic signatures at stem heights of 0.1 and 8 m, we optimized FPLDs that can be described by the following parametric distribution:

$$h_{\text{F}}(s; \alpha, \beta, \gamma) = F\left(\frac{s - \gamma}{\beta}; \alpha, 100\right) \qquad (22)$$

with $F$ being the probability density function of the Fisher–Snedecor distribution. the second parameter of $F$ was set to a fixed value of 100, so that its first parameter $\alpha$ acts as shape parameter while the additional parameters $\beta$ and $\gamma$ act scale and lag parameters, respectively.

Apart from simply predicting the transformation of $\delta_{\text{RWU}}$ during its transmission through a tree's xylem, the fitted FPLDs were also used to infer time variable xylem water age distributions. This was achieved by applying the approach described in Sec. 2.1.3 in combination with a series of virtual tracers, one for each time step with a concentration of 1 during the respective time step and 0 during all other time steps. The resulting tracer concentration time series were used to infer xylem water ages similar to Sprenger et al. (2016) (ages of percolation water below the root zone) and Brinkmann et al. (2018) (ages of RWU water). In contrast to the two mentioned studies, the water ages in this study refer to the time of tree water uptake, instead of the time of input as precipitation into the system.

## 3 Results

### 3.1 Soil moisture and sap flow

The temporal dynamics of daily sap flow velocities and soil moisture (averaged across the two profiles on the irrigated plot) are depicted in Fig. 4a & b. Averaged for each month, the mean daily sap flow velocities varied between 42, 88, 94 and 85 cm/d for May (starting at 21 May), June, July and August (ending at 8 August), respectively.

The sprinkling experiment started on 22 May during a wet period with soil moisture around field capacity (ca. 30 Vol.%). From 1 June to 10 June, the first dry period occurred, with soil moisture in the topsoil decreasing towards 15 Vol%. Several rain events between 10 June and 15 June rewetted the soil, but afterwards the soil moisture in all depths declined considerably. One rainfall event with 29 mm on 12 July caused a brief rewetting of the topsoil and a heavy convective event with 52 mm on 27 July bypassed the upper two soil moisture probes and caused a strong increase by around 10 Vol.% at 40 and 60 cm depth.





**Table 2.** The different parameters used for RWU modelling and the transfer functions used to transform $\delta_{\mathrm{RWU}}$ into $\delta_{\mathrm{R}}$ ($h_{\mathrm{R}}$, $h_{\mathrm{F1}}$) and $\delta_{\mathrm{R}}$ into $\delta_{\mathrm{H}}$ ($h_{\mathrm{F2}}$). Meanings of the parameters are listed in Table A1.

| Parameter | model | optimization range | final value |
|---|---|---|---|
| $\theta_w$ | RWU | 5–10 % | 8 % |
| $\theta_s$ | RWU | 45 % | 45 % |
| $\overline{\theta_{c1}}$ | RWU | 10–90 % | 40 % |
| $\overline{\theta_{c2}}$ | RWU | 100 % | 100% |
| $z_{\mathrm{r}}$ | RWU, $h_{\mathrm{R}}$ | 0.4–2 m | 0.9 m |
| $r_{\max}$ | $h_{\mathrm{R}}$ | 1–5 m | 3 m |
| $\lambda$ | $h_{\mathrm{R}}$ | 0.1–20 | 20 |
| $\alpha_1$ | $h_{\mathrm{F1}}$ | 1–30 | 3.4 |
| $\beta_1$ | $h_{\mathrm{F1}}$ | 0.5–3 | 1.43 |
| $\gamma_1$ | $h_{\mathrm{F1}}$ | 0 m | 0 m |
| $\alpha_2$ | $h_{\mathrm{F2}}$ | 0.1–3 | 0.62 |
| $\beta_2$ | $h_{\mathrm{F2}}$ | 0.5–3 | 1.3 |
| $\gamma_2$ | $h_{\mathrm{F2}}$ | 0–5 m | 1.64 m |

The fitted Eq. 2.3 established a solid relationship ($R^2$ of 0.85) between $v_{\mathrm{sap}}$ and VPD (see Fig. A4a). Since there are no systematic discrepancies between measured and VPD-derived $v_{\mathrm{sap}}$ values (see Fig. A4a), we have high confidence in assuming

that the observed periods of reduced $v_{\mathrm{sap}}$ during June and July (see Fig. 4a) can be attributed to temporarily lowered VPD instead of soil water deficits.

### 3.2 Stable water isotopes

In each measurement sequence 15 probes (2 standards, 6 SWIPs and 7 XWIPs) were measured within 3 to 5 hours, resulting in one measurement every 12 to 20 minutes. The raw aggregated measurements and the correction procedure are described in

Appendices A1 and A2. Some short term fluctuations in the high frequency data caused by diel air temperature fluctuations could be observed that could not completely be removed by the postprocessing procedures. Due to the large amount of data points in the full data set and our interest in the overall temporal dynamics, we limited our further analysis to daily median values of the full data set, setting aside the development of a more robust diel calibration procedure for future studies.

Three of the xylem probes (T1B, T2B and T3B) exhibited a negative $\delta^{18}$O bias, which was corrected as described in

Appendix A2.2. A comparison of the probe heads right after removal from the stem (12 weeks after installation) revealed that the membrane heads of two $\delta^{18}$O biased probes were covered by biofilms while an unbiased probe did not show such a biofilm (see Fig. A6).

Focusing on the first period, between 21 May (date of the irrigation) and 10 June, the soil $\delta$D-signature was rather stable, while $\delta_{\mathrm{xyl}}$ increase for 6 to 14 days to reach a plateau. The $\delta$D signatures at the stem base at 10cm height (green triangles in





Fig. 4 c & d) showed the steepest rise and reached their plateaus after approximately 6 days while the probes installed at 8 m height (pink triangles in Fig. 4) showed a delay of around 14 days. The $\delta$D signatures at 1.5 m tree height (yellow symbols) responded in-between the former two groups.

The immediate post-irrigation $\delta^{18}$O signatures of the soil profile showed no considerable depth differentiation and little dynamics (see Fig. 4c). Following some smaller precipitation events with elevated $\delta^{18}$O signatures (-7.2 ‰ in rain compared

to -9 ‰ in the soil), the soil signatures slightly shifted upwards. A similar development can be observed for the xylem $\delta^{18}$O signatures following the soil signatures. The larger rainfall event in the night between 10 June and 11 June, followed by another 13 mm event on 15 June influenced the soil isotopic signatures in two ways: the upper 20 cm of the soil were enriched in $\delta^{18}$O (2 to 3 ‰ more enriched than the lower soil depths) and the $\delta$D enriched label water introduced with the irrigation was percolating further downwards. Following those changes in the soil isotopic signatures, we see the xylem $\delta^{18}$O signatures

increasing by about 1.5 ‰ within 5 to 10 days. Simultaneously the xylem $\delta$D signatures decrease to levels close to the topsoil $\delta$D signatures.

We observed two periods 23 June to 11 July and 16 July to 26 July that are characterized by declining soil moisture and rather constant $\delta_{\text{soil}}$. Both of these periods show the same pattern of $\delta_{\text{xyl}}$ further deviating from the isotopic composition of the topsoil and converging towards the values of the deeper soil layers. On the other hand, we also observed two rainfall events

(12 July and 27 July) that lead to a replenishment of soil moisture without considerable changes in $\delta_{\text{soil}}$. In both cases, we could see an opposite response to the dry period. $\delta_{\text{xyl}}$ was more similar to the isotopic composition of the upper soil layers and diverged from that of the deeper soil layers. The 18 mm of rainfall on 12 July were exhausted within the following days and soil moisture (and $\delta^{18}$O signatures) quickly returned the to the low levels before the event. The rainfall event on 27 July raised the soil moisture levels to such an extent that the low pre-event soil moisture did not occur again within the observed

time period.

### 3.3 Optimization of the RWU model

The temporally and spatially continuous soil moisture and soil water isotope data needed for the optimization of the RWU model was obtained by interpolation of soil sensor data, soil core measurements and SWIP measurements (see Fig. A5).

Subsequently, the optimization was carried out as described in Sec. 2.3.1. Figure 5 shows the results for the combined

parameter optimization of the RWU model and the transfer function $h_{\text{R}}$, which is used to convolve the modelled $\delta_{\text{xyl.RWU}}$ in order to obtain values for validation of $\delta_{\text{xyl.R}}$. Each row in Fig.5 belongs to one model output variable ($\delta^{18}$O, $\delta$D and the RWU depth distribution index $r_U$) and was evaluated individually. The best ten model simulations are depicted in blue, 20 random samples drawn from the worst 50–90% of all simulations are depicted in red/orange. The evaluation was done for the starting phase (dark blue and red) and for the full observation period (light blue and orange).

When only the first half of the observational record is considered (darker blue squares and lines), both isotopes show a parameter optimum for $z_{\text{r}}$ (rooting depth parameter) between 0.75 m and 1 m (see Fig. 5(d&h)). For that period, it was not possible to identify the optimal parameter value for the other two (water stress related) RWU parameters $\theta_w$ and $\overline{\theta}_{c1}$, which seemed to be rather insensitive to both isotopes (see dark blue squares in Fig. 5(e,f,i&j)).





The maximum lateral root extent $r_{max}$ is critical to reproduce the rise of the Deuterium signal after the irrigation and showed
a clear optimum around 3 m (Fig. 5k). The lateral root density decay parameter $\lambda$ proved to be unidentifiable and was omitted
in Fig. 5. When the full time period was considered (light blue diamonds and lines), the optimal values for $z_r$ were found at
deeper depths between 0.9 and 1.5 m (see 5(d&h)). The optimal wilting point soil moisture $\theta_w$ ranges between 6–8 % and 7–9
% (for $\delta^{18}O$ and $\delta D$, respectively, see Fig.5(e & i)). Once again, optimal values for $r_{max}$ are at around 3 m, this time for both
isotopes.

Unlike of the two water isotopes, the water uptake ratio $r_{RWU}$ relies on soil moisture data alone and is not involved in the
convolution step. Consequently, $r_{max}$ was completely insensitive to $r_{RWU}$. With respect to $r_{RWU}$, optimal $z_r$ values are found
between 80 and 100 cm, which is in good agreement with the isotope based $z_r$ values in the start phase of the observational
record. Based on $r_{RWU}$, the optimal $\theta_w$ values are between 7 and 9 % (see Fig.5m) and the optimal $\overline{\theta}_{c1}$ values are between 30
and 100 % (see Fig. 5n).

### 3.4 Optimization of xylem FPLDs

Based on the optimized RWU model, we could compare modelled $\delta_{RWU}$ to measured $\delta_{xyl}$ values. Except for the Deuterium
signatures towards the end of the Experiment, there was a good agreement for both isotopes (see Fig. 6). However, in cases
of abrupt $\delta_{RWU}$ changes, the observed $\delta_{xyl}$ values did respond with a delay which increased with stem height. This was most
obvious after the Deuterium-labelling (box A in Fig. 6b). In order to account for the expectable delay that occurs during the
transport of water from the roots along the xylem, we optimized FPLDs to transform our modelled $\delta_{RWU}$ values into $\delta_{xyl}$ values.

Figure 7 shows observed (squares) and modelled (lines) $\delta_{RWU}$ (grey) and $\delta_{xyl}$ (green and pink) values. The original time
series was transformed into the sap distance domain (see Sec. 2.1.3) in order to eliminate the influence of time variable $v_{sap}$,
which was lower at the start of the depicted period and higher towards its end. Figure 7(b) depicts the same data as Fig. 7(a),
but plots their rate of change instead of their absolute values. The colored lines in Fig. 7(a&b) are the results of convolutions
of the modelled $\delta_{RWU}$ signature with different FPLDs (depicted in Fig. 7(c).

In order to transform $\delta_{RWU}$ into $\delta_{xyl.R}$, we convolved it with two different transfer functions: (1) the conceptually grounded
$h_R$ (thin green line, defined in Sec. 2.1.2), whose parameters were optimized together with the RWU model parameters and (2)
the empirically, more flexible $h_{F1}$ (thick green line, defined in Eq. 22), whose parameters were optimized subsequently to reach
the best possible fit to observed $\delta_{xyl.R}$ values. The overall shapes of $h_R$ and $h_{F1}$ turned out to be similar, but the latter achieved
a much better fit to the available $\delta_{xyl.R}$ observations, while the former failed to adequately reproduce the right-skewed, tailed
shape required to fit the observational data (see Fig. 7(b)).

Following this, we simulated the signal transformation of $\delta_{xyl.R}$ (at the stem base, thick green line) into $\delta_{xyl.H}$ (at 8 m stem
height, thick pink line) by convolving $\delta_{xyl.R}$ with another transfer function, $h_{F2}$ (dashed pink line in Fig. 7(c)), whose parameters
were optimized to reach the best possible fit to observed $\delta_{xyl.H}$ values. In contrast to $h_{F1}$, $h_{F2}$ features a notable time lag around
1.4 m at the beginning and has a strong peak. It's tailing is similar to that of $h_{F1}$.

To directly transform $\delta_{RWU}$ into $\delta_{xyl.H}$, it can be convolved with $h_{F1F2}$, which is the convolution of the two FPLDs $h_{F1}$ (root
tips to stem base) and $h_{F2}$ (stem base to 8 m stem height).





According to the shape of $h_{F2}$, the largest fraction of a signal between the stem base and a stem height of 8 m arrives after 1.4–2 m of cumulative sap flow distance. This may seem paradox, but it is not. The sap flow distance $s$ is derived from heat probe

based $v_{sap}$, which other studies (Meinzer et al., 2006; Schwendenmann et al., 2010) have reported to be considerably lower than $v_\delta$ (transport velocity inferred from isotopic tracer observations). Based on our observations, we can infer $v_\delta$ between the stem base and 8 m stem height to be about 5.5 times faster than $v_{sap}$.

### 3.5 Xylem water age distributions

In order to compute temporally variable distributions of xylem water ages at a certain stem height, two things were required:

(1) a transfer function representing the FPLD between root tips and the stem height of interest, i.e. $h_{F1}$ and $h_{F1F2}$ as determined within the previous section, and (2) a sap flow time series.

We extended our measured sap flow time series beyond its original range over a full year as described in Sec. 2.3 (see Fig. A4b for the complete extended $v_{sap}$ time series). Subsequently, we duplicated this extended $v_{sap}$ time series in order to obtain enough data for an appropriate warm-up period for the following computations.

By combining sap flow velocities, FPLDs, and virtual tracers for each day (as described at the end of Sec. 2.3.1), we obtained xylem water age distributions at 0.1 and 8 m stem height. In Fig. 8 the time variable age distributions are represented through specific quantiles between 0 and 99% for 8 m stem height. For clarity's sake, only the median water age is depicted for 0.1 m stem height .

Figure 8(a) shows the xylem water ages over the course of a year between the ends of two vegetation periods. During the

dormant season (November – May) xylem water is immobile and consequently it's age is increasing by one day per day, eventually exceeding ages of 200 days and more. At the start of the growing season, xylem water ages drop sharply, as soon as fresh uptake water has replaced the previous season's water, which happens earlier at the base of the tree stem.

Figure 8(b) focuses on the the growing season of our field experiment. At the start and towards the end of the growing season, xylem water ages are considerably larger than during the main growing season. From the beginning of June to mid

of September the median xylem water age at 8 m stem height varies between 2.3 and 8 days with a mean value of 4.7 days. Median xylem water ages at 0.1 m stem height in the same period are lower and range between 0.4 to 3.8 days with a mean value of 1.6 days.

## 4  Discussion

### 4.1  Performance of the measurement setup

By complementing the measurement system of Volkmann and Weiler (2014) with a sophisticated hard- and software framework to control the required gas flow controllers and solenoid valves, we developed a system capable of largely unattended long-term operation. As all of the additional components were built from readily available parts, our extension of the original setup did



not notably increase the overall cost, which is mainly set by the CRDS itself and to a smaller degree by the required probes and valves.

Up to this date, our setup is the most complete field tested in-situ stable water isotope measurement system for continuous ecohydrological investigations. It contains many of the elements of the "ideal system" sketched by Beyer et al. (2020). To summarize, we list the main advantages of the use of WIPs compared to other in-situ measurement techniques:

1.  To install WIPs into the soil, it suffices to dig or drill a hole with a diameter of around 30 cm. Multiple WIPs (in different depths, at different sectors of the hole) can then be pushed into practically undisturbed soil. The installation of loops of
495        gas permeable tubing into the soil, as used by Rothfuss et al. (2013), Oerter and Bowen (2017) and Kübert et al. (2020), is much more invasive and will inevitably disturb the observed soil to a higher degree.

2.  The borehole equilibration method by Marshall et al. (2020) relies on boreholes that go all the way through the tree stem. Consequently, with increasing stem diameters, the measurements of this method will increasingly be influenced by the isotopic signature of immobile water from the heartwood. Possibly up to a point where the dynamics of the mobile water
500        transported in the outer parts of the xylem gets hard to detect. WIPs on the other hand are always probing the outer 5 cm of the stem xylem - the same depth as measured with many sap flow sensors.

3.  By directly diluting the sample gas within the WIP's mixing chamber, condensation within the sample line is avoided without the need for additional heating of the whole sample line. Additionally, there is an easy way to test the airtightness of the measurement system, by checking how dry a highly diluted sample gets - if it remains moist at maximum dilution
505        rate, there must be some source of moisture (i.e. condensed droplets) between mixing chamber and analyser or other leaks.

4.  The identical design of WIPs in soil and xylem, provides a consistent measurement method for soil and tree xylem. This can be considered as an advantage compared to in-situ methods that are only suited for soil (Rothfuss et al., 2013) or xylem (Marshall et al., 2020). The use of one single type of probe simplifies automating the measurement procedure and
510        interpreting the obtained measurements.

We encountered some issues with partially biased $\delta^{18}$O XWIP measurements, similar to what Volkmann et al. (2016b) have reported. Since not all XWIPs exposed such a $\delta^{18}$O-bias, we hypothesize that the occurrence of the encountered bias is related to the observed formation of biofilms on the XWIP probe heads (see Fig. A6). However, even when the $\delta^{18}$O measurements seemed biased, they exposed similar temporal dynamics as unbiased measurements and all of those dynamics agreed with
what our proposed modelling framework predicted. So even though we may not be completely sure how to reliably rule out the formation of $\delta^{18}$O-bias inducing biofilms, we are confident that the observed temporal dynamics will still contain valuable information. If unbiased xylem measurements or observations of soil isotopic data are available, a bias correction is always possible.

Considering the expected spatial heterogeneity of a skeleton rich, clayey soil – on top of the spatial heterogeneity of infil-
tration patterns within forest stands (Goldsmith et al., 2019) – and the relatively slow temporal dynamics of the observed soil





isotopic signatures, future investigations might benefit using an increased number of WIP soil profiles at the cost of a lower temporal resolution of soil isotope measurements.

## 4.2    Standards and calibration

Our standard probes were sampling the vapour from the headspace of sealed containers filled with waters of known isotopic

composition. This contrasts to other practices of using soil standards (Beyer et al., 2020), prepared from dried soil material that was rewetted with standard water. The main reason of using headspace standards was the situation, that the maximum number of WIPs in our setup was limited and that such soil standards may not be representative for xylem isotope measurements. Furthermore, the sampling of field soil for a "representative" soil standard is problematic when soil properties vary with depth – a standard suited for one horizon may lead to biased results for another horizon. Furthermore, Gaj et al. (2017) have shown

that clay minerals may lead to isotopic fractionation when a label water is applied to oven dried clay rich soils. This might lead to biases that are difficult to attribute to the different samples and hence liquid water standards may be the better choice.

For a WIP with fixed dilution rate placed in the headspace of a liquid water standard with variable temperatures, we found a close relationship between the sample gas' vapour concentration and isotopic composition (see Fig. A2). As long as the air entering our probes is vapour saturated (which we assume to be the case within transpiring trees and not overly dry soils),

the sample gas vapour content is directly related to the temperature of the sampling location. Therefore, we ended up with a calibration procedure that uses the sample gas vapour concentration, which is automatically measured by the CRDS, instead of measured temperatures at the sampling locations. When all measurements of one probe per day were averaged, the resulting time series were reasonably consistent on a day-to-day scale. On a shorter timescale we observed fluctuations (for xylem- and soil probes, but with clearly more pronounced fluctuations for soil probes closer to the surface), which were likely artefacts

of insufficiently compensated temperature effects. A calibration procedure that explicitly considers observed temperatures at each WIP might help to dissolve these sub-daily fluctuations and therefore improve the measurement accuracy and enable the exploitation of the full temporal potential of these high frequency in-situ measurements. But the proper consideration of temperature effects is difficult under field conditions, especially when considering tree stems, where strong temperature gradients can occur at small spatial extends (see (Derby and Gates, 1966)). Even for laboratory conditions, Rothfuss et al.

(2013) reported a mismatch between their measurements and the Majoube (1971) equation to compute the isotopic fractionation between the liquid and the vapour phase under equilibrium conditions as a function of temperature.

## 4.3    Process based RWU modelling

We were able to identify meaningful parameters ($z_r$, $\theta_w$ and $r_{max}$) for our process-based RWU model. This model in turn was suited to predict $\delta_{RWU}$. Our approach of driving the RWU model directly with (interpolated) measured data avoided the

difficulties that are likely to occur during the simulation of water transport within highly structured, skeleton rich soil. Yet, our data driven approach prevented problematic parametrizations of the RWU model from accumulating errors over consecutive time steps as it would happen when the model itself had to keep track of soil moisture and soil water isotopic composition.





Consequently, the critical model parameters $\theta_w$ (wilting point) and $\overline{\theta}_{c1}$ (onset of uptake reduction) were not identifiable ($\theta_w$ was insensitive when optimized against isotope data, but identifiable when optimized against soil moisture).

Regarding the RWU $\delta$D signatures, there is a notable mismatch for the second half of our observation period: the model systematically predicts more enriched $\delta_{RWU}$ values than we could measure at $\delta_{xyl}$. Since all of the measured soil $\delta$D values during that period were more enriched than the observed $\delta_{xyl}$, this is not a failure of the model itself, but rather a consequence of insufficient model input data. This could hint to at deeper uptake depths which were not covered by the SWIP depth profile, the general lack of representativeness of one single SWIP profile due to soil heterogeneity or in the worst case a systematic

bias in XWIP $\delta$D measurements as an aftereffect of the 10 day measurement interruption in July.

## 4.4    Interpretation of FPLDs

Unlike most hydrological catchments, where FPLDs change with the length of the flowing stream network (Van Meerveld et al., 2019), the tree xylem may be considered to feature a static FPLD which is constant irrespective of the occurring transport velocities. Consequently, xylem water ages are determined by a static FPLD and time variable tracer transport velocities ($v_\delta$).

While it is nearly impossible to monitor $v_\delta$ with a high temporal resolution, $v_{sap}$ can be used as a proxy for $v_\delta$.

    Previous experiments have shown that $v_\delta$ may exceed $v_{sap}$ by several hundred percent. Meinzer et al. (2006) reported $v_\delta$ to be five times higher than $v_{sap}$ and Schwendenmann et al. (2010) even reported it to be about 16 times higher than $v_{sap}$. This can be explained by the fact that not all of the xylem area probed for $v_{sap}$ measurements is actively contributing to sap transport. Unaccounted wounding errors from the probe installation may lead to additional underestimations of $v_{sap}$. Irrespective of a

possible systematic scaling error, Meinzer et al. (2006) reported a good correlation between $v_{sap}$ measurements $v_\delta$ values. We also observed a mismatch between $v_{sap}$ and $v_\delta$ : $h_{F2}$, the optimized FPLD between 0.1 m and 8 m height, featured a lag of only 1.43 m (in the sap distance domain based on $v_{sap}$), which indicates, that the actual FPLDs are likely 5.5 times longer than our apparent, $v_{sap}$ based FPLDs shown in Fig. 7(c). However, this scaling error is of no consequence as long as the apparent FPLDs are not interpreted directly, but simply used to predict tracer time series.

Our conceptually derived $h_R$, which was based on assumed vertical and lateral root distributions, proved suited to act as FPLD between $\delta_{RWU}$ and $\delta_{xyl.R}$ Yet, due to a lack of observations of lateral root distributions, we had to estimate an appropriate value for the maximum lateral root extent $r_{max}$ via model parameter optimization, where $r_{max}$ was identifiable when the model was evaluated with respect to $\delta_{xyl.R}$. Given the uncertainties arising from measurement precision and accuracy, as well as spatial heterogeneity of the soil, it turned out that the robustness of an $r_{max}$ estimate greatly improves with a distinct artificial isotopic

labeling pulse.

    However, the overall shape of the conceptually derived $h_R$ was unable to fully reproduce the skewness and tailing of the fitted parametric distribution $h_{F1}$ (from Eq. 22), which produced an even better fit to the observed data. The observed tailing might be a consequence of an unexpected lateral root distribution, but it is more likely to be be caused by dispersion during the the root xylem passage, as we see a similar tailing for the transport along the stem (represented by $h_{F2}$). Future labeling

experiments with sequentially applied tracer pulses limited to certain radial distances from the stems of trees with different characteristics could help to study macroscopic and microscopic factors that shape the FPLD of a tree's root xylem.





### 4.5 Xylem water ages, FPLDs and their implications

During the main growing season 50% of all xylem water passing the stem base of our studied beech trees was older than 1.6 days. When looking at the 8 m stem height, the mean median xylem water age was 4.7 days. Assuming the xylem water transport velocities are not changing fundamentally above that point, we can estimate that the mean median water age of the xylem water within the tree crowns of our studied trees was close to 10 days. Towards the fringes of the growing season, we can expect considerably higher water ages. This has to be kept in mind for any type of xylem water sampling in order to investigate RWU, specifically after rainfall events or artificial isotopic labelling, but also during periods where water stress may lead to changes in $\delta_{RWU}$.

Previous studies (James et al., 2003; Meinzer et al., 2006; Schwendenmann et al., 2010; Gaines et al., 2016) have typically investigated above ground xylem water transport via injection of isotopic tracers into the stem and they gathered valuable data regarding the transport processes between the stem base and crown branches or leaves. However, by injecting their tracers above the ground, they did not capture the below ground component of xylem water transport, between all individual root tips and the stem base.

Knighton et al. (2020) citing the study of Gaines et al. (2016), claimed they had "estimated time lags between RWU and transpiration", while they actually had estimated time lags between tracer injection to the stem base and transpiration. Similarly, de Deurwaerder et al. (2020) propose a modelling framework that erroneously equates $\delta_{RWU}$ to $\delta_{xyl}$ at the stem base, leading them to the prediction of unlikely clear $\delta_{xyl}$ signals further up the stem. The FPLD between root tips and stem base ($h_{F1}$ in Fig. 7(c)) contributes considerably to the smoothing of $\delta_{RWU}$, much more than the FPLD along the first 8 m stem height ($h_{F2}$ in Fig. 7(c)). Consequently, we strongly suggest to include the below ground fraction of the tree into any endeavour that aims to simulate the propagation of $\delta_{RWU}$ along the stem.

At that point, it is important to note that most of the root system FPLD's distribution form results from its spatial configuration, featuring not only a depth density distribution, but also a lateral abundance (lateral density combined with projected area) distribution. Nevertheless, the often used simplified vertical 1-D representation of the soil-plant system for RWU investigation purposes does not necessarily have to be expanded by a second dimension: a computationally efficient way to account for vertical and lateral root distributions can be achieved by convolution of $\delta_{RWU}$ with an appropriately shaped FPLD.

## 5 Conclusions

This study demonstrated the application of a measurement setup which facilitates unprecedented high frequency monitoring of stable water isotopes in soil and tree stem xylem. We were able to predict the observed time series of xylem water isotopic concentrations at different stem heights with a combination of a process based RWU model and a convolution based approach that accounts for the FPLD between root tips and the sampling points in the tree stem.

Our results showed, that $\delta_{RWU}$ and $\delta_{xyl}$ are often similar but not necessarily the same. No $\delta_{xyl}$ measurement can represent actual $\delta_{RWU}$: it will always be an integration over certain fractions of $\delta_{RWU}$ from different points in the past. Therefore, we have to reject our first hypothesis, that a tree's xylem water isotopic signature is equivalent to the isotopic signature of RWU.



Only under certain conditions (i.e. little temporal variability of the RWU composition or short transport distances combined with high xylem water transport velocities) can FPLDs within the plant safely be neglected. Otherwise, plant water isotope models should account for the FPLDs between RWU and the sampling point used for model evaluation or between RWU and transpiration in the leaves.

    Regarding our second hypothesis, we conclude that a conceptual representation of a tree's root system is capable to reproduce

the basic shape of the FPLD between $\delta_{RWU}$ and $\delta_{xyl}$. But more detailed observations and experiments are needed before tree root xylem FPLDs can robustly be derived from observable tree characteristics and the tree architecture.

    Due to the smoothing effect of FPLDs, sub daily observations of tree xylem water isotopes are unlikely to reveal much information about short term RWU dynamics. Except for precipitation events, soil water isotopes show even smaller temporal dynamics. Consequently, future investigations of stable water isotope dynamics should favour an increased number of probes

in soil and xylem to cover spatial heterogeneity over high subdaily measurement frequencies.

*Code and data availability.* The supplement to this study contains the processed WIP measurements and all the climate-, sap flow- and soil moisture data needed to reproduce the essential results presented in this article. Furthermore, the supplement contains R-scripts that recreate the 2-D interpolation of soil moisture and isotopes (*S1_soilInterpolation.r*), the computation of $\delta_{RWU}$ (*S2_XWIPs_and_RWU.r*), the time series transformation and convolution of isotopic data (*S3_transformation_and_convolution.r*) and the computation of xylem water age

distributions (*S4_xylem_water_age.r*).

**Appendix A**

**A1    Raw in-situ measurements**

Fig. A1 shows the aggregated (averaged last two minutes before valve switch) CRDS raw data for both stable water isotopes (Fig. A1a&b) as well as the sample gas moisture content (Fig. A1c). The measurement sequence including 15 probes was

completed each 3 to 5 hours, resulting in one measurement every 12 to 20 minutes. The standards and the xylem probes showed a greater temporal variability resembling diurnal temperature fluctuations, while the thermally better insulated soil probes yielded more constant values over the day. Since the high frequency data seems to be dominated by those temperature related diurnal fluctuations and there are so many data points, we chose to focus on daily median values, which are plotted solid in Fig. A1, while the underlying data points are plotted transparently in the background. On top of Fig. A1 we indicate

some incidents that need to be commented on:

    (A) We missed to measure the initial conditions and our first measurement days might have been impaired by moisture within our measurement system that first had to be flushed out over time.

    (B) During a rainfall event, stem flow intruded along the shaft of an insufficiently sealed WIP (T1R) diagonally installed into a tree root. After the water had entered the system around early 11 June, subsequent measurements of all other probes

were unusable until we managed to visit the field site at June 13 to manually flush all tubing with dry pressurized air. Then we





**Table A1.** Abbreviations and parameter symbols used within this article and their meanings.

| Abbreviation / Symbol | Description |
| --- | --- |
| FPLD | Flow path length distribution |
| RWU | Root water uptake |
| SWIP | Soil water isotope probe |
| WIP | Water isotope probe |
| XWIP | Xylem water isotope probe |
| $r_{RWU}$ | Ratio of water uptake from upper soil layers to water uptake from lower soil layers |
| $\delta_{RWU}$ | Isotopic signature ($\delta^{18}$O and $\delta$D) of root water uptake |
| $\delta_{soil}$ | Isotopic signature ($\delta^{18}$O and $\delta$D) of soil water |
| $\delta_{xyl}$ | Isotopic signature ($\delta^{18}$O and $\delta$D) of xylem water |
| $\delta_{xyl.R}$ | Isotopic signature ($\delta^{18}$O and $\delta$D) of xylem water at the stem base |
| $\delta_{xyl.B}$ | Isotopic signature ($\delta^{18}$O and $\delta$D) of xylem water at 1.5 m stem height |
| $\delta_{xyl.H}$ | Isotopic signature ($\delta^{18}$O and $\delta$D) of xylem water at 8 m stem height |
| $v_{sap}$ | Sap flow velocity (as estimated with heat pulse probe) |
| $v_{\delta}$ | Tracer transport velocity (as observed from labelling pulse) |
| $\theta_w$ | Volumetric soil water content at wilting point |
| $\theta_s$ | Volumetric soil water content at saturation |
| $\overline{\theta_{c1}}$ | Normalized volumetric soil water content at onset of water stress (see Eq. 4) |
| $\overline{\theta_{c2}}$ | Normalized volumetric soil water content at onset of aeration stress (see Eq. 4) |
| $z_r$ | Root depth above which 95 % of roots can be expected (see Eq. 3) |
| $r_{max}$ | Maximum lateral extent of the rooting system (see Eq. 6) |
| $\lambda$ | Lateral root density decay parameter (see Eq. 6) |
| $\alpha$ | shape parameter for transferfunction $h_F$ (see Eq. 22) |
| $\beta$ | scale parameter for transferfunction $h_F$ (see Eq. 22) |
| $\gamma$ | lag parameter for transferfunction $h_F$ (see Eq. 22) |

renewed the silicone sealing to protect probe TR1 from further water intrusions. While all other probes were back measuring, probe TR1 needed five more days to return to its normal operation.

(C) During a storm event, some cable connections became loose. Starting with probe T1H, and a few days later continuing with the neighbouring probes T1R and (SWIP) 100 cm. The valves for those three probes did not switch and the CRDS's vacuum pump sucked its sample gas through the least air tight point in our assembly, rendering the respective data useless.

(D) Some time after changing the gas cylinder holding the dry air at 2 July, a seal failed - emptying the gas cylinder much faster than expected. Due to a holiday leave it was not fixed before 11 July.

(E) Some animal nibbled through the tubing at the shaft of the xylem probe T2R and severed the tubing completely from the probe.





## A2   Post-processing of in-situ measurements

### A2.1   H$_2$O-ppmV correction

Following the volumetric moisture correction procedure described in section 2.3, the relationship between volumetric sample moisture and the raw vapor $\delta^{18}$O and $\delta$D for both standards are shown in Fig. A2(a & b), respectively. We split the whole data set into shorter periods (as indicated in Fig. A2(c & d) in order to check the consistency of this correction procedure. Two periods are striking: the first period (2019-05-22 to 2019-06-01) exhibits a lot of scatter for the heavy standards' $\delta^{18}$O values and also shows a strikingly high slope for the $\delta$D values. The second striking period is the fifth period (2019-07-03 to 2019-07-11) and coincides with the interrupted gas supply described in section 3.1.4. Here, the slopes for $\delta^{18}$O are higher and there is a clear negative offset of the $\delta$D values compared to all other periods. Over all of the seven sub periods and both standard probes, the slope for $\delta^{18}$O ranges from $1.59 \times 10\text{-}4$ to $2.43 \times 10\text{-}4$ with a mean of $1.94 \times 10\text{-}4$ (all values in ‰/ppmV) and the slope for $\delta$D lies between $3.84 \times 10\text{-}4$ and $1.04 \times 10 5$ with a man value of $6.26 \times 10\text{-}4$ (also all values in ‰/ppmV). Despite this notable spread of slopes, it turned out that the differences between the utilization of one (overall mean) slope for the moisture correction over the whole experiment does not lead to big differences compared to the utilization of period specific slopes.

### A2.2   Manual data corrections

In terms of $\delta^{18}$O, three XWIPs (T1B, T2B and specifically T3B) exhibited similar dynamics in their time series compared to the remaining XWIPs (T1R, T1H, T2R and T2H), but seemed to have a negative offset, without any corresponding offset in their $\delta$D time series (Fig. A1(a & b). Fig. A3(a) shows all measurement values after moisture correction and normalization to the liquid phase. Assuming no further sources of water uptake than the observed soil profile and no fractionation during root water uptake, all of the xylem measurements (green, yellow and pink symbols) in Fig. A3(a) should be located within the green triangle of possible mixtures of soil water signatures (blue circles). The pink shaded area roughly marks the plot region containing implausible (i.e. impossible to achieve by soil water mixing) xylem water signatures. The values of T2R below the global meteoric water line (right half of A3(a)) fall into the time with interrupted gas supply (see Fig. A1) and were completely discarded. The XWIP values at the lower centre of the plot fall into the starting period of the time series and are close to the values measured within the soil cores taken before the irrigation. The XWIP values to the left are the ones earlier mentioned (same dynamics, negative $\delta^{18}$O offset). We shifted these values into the plausible range by manually offsetting their $\delta^{18}$O time series until they fell into our mixing triangle. For T1B we added 2.5 ‰ to all values, for T2B we added 1.5 ‰ starting at 1 June and another 0.5 ‰ starting from 11 July. For T3B we added 3.75 ‰ to all values and additional 2 ‰ starting at 22 June.



**Figure 4. (a)** Daily rainfall and cumulative sap flow distances. Dark blue squares represent the mean rainfall amounts collected with the four bulk samplers (dark blue lines indicating the periods over which the bulk samples were collected). **(b)** Volumetric soil moisture at 4 soil depths. **(c & d)** Corrected time series of $\delta_{soil}$ and $\delta_{xyl}$ in liquid water. The blue squares represent the mean isotopic signatures of the four bulk samplers. The translucent grey areas in (c & d) mark time periods with missing or unreliable isotope data.

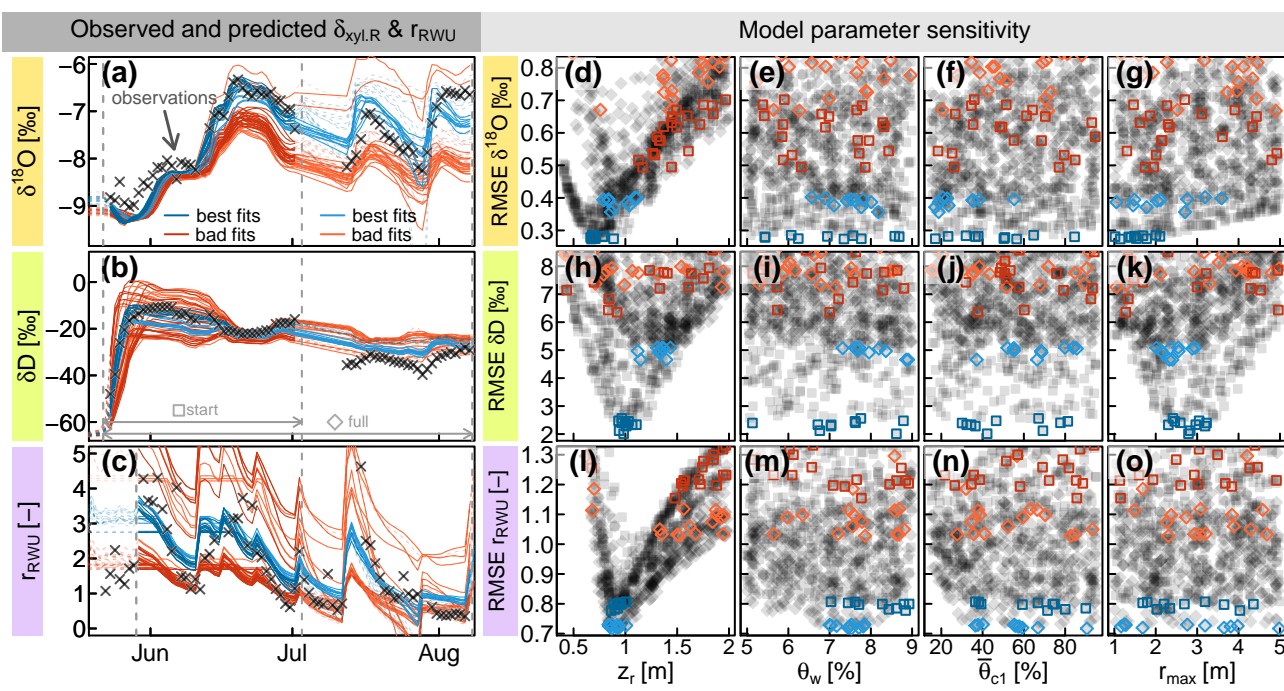

**Figure 5. (a & b)** Observed $\delta_{xyl.R}$ ($\delta_{xyl}$ at the stem base, grey crosses) compared to model predictions of $\delta_{xyl.R}$ (lines), evaluated over the whole observational record (light blue and orange) or just the start phase (dark blue and red). **(c)** root water uptake ratio $r_{RWU}$ derived from soil moisture measurements (grey crosses) and RWU simulations (coloured lines). **(d–o)** RMSE values between observed signatures and simulated signatures in relation to the model parameters $z_r$ (rooting depth), $\theta_w$ (wilting point soil moisture) $\overline{\theta}_{c1}$ (critical normalized soil moisture of water stress onset) and $r_{max}$ (maximum lateral root extent). The parameter combinations leading to the selected coloured time series in panels a–c are coloured identically in the respective panels (d–o)



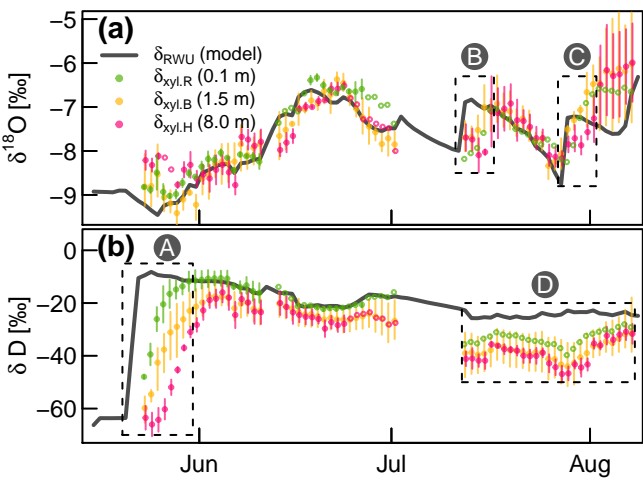

**Figure 6.** Measured $\delta_{xyl}$ (coloured dots) compared to the modelled $\delta_{RWU}$ (grey line). $\delta_{xyl}$ values were averaged across all available probes at each of the three installation heights. Vertical bars indicate the value range of all available (2–3) probes, empty circles without lines indicate single probe values. Dashed boxes A, B & C: periods, where observed $\delta_{xyl}$ apparently lags behind changes in modelled $\delta_{RWU}$. Dashed box D: clear bias between modelled $\delta_{RWU}$ and measured $\delta_{xyl}$.

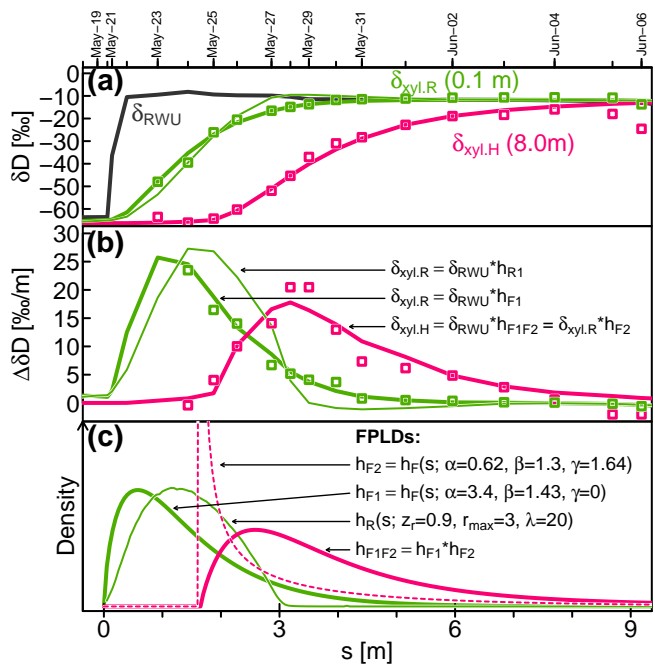

**Figure 7.** Measured (squares) and modelled (lines) isotopic signatures of RWU($\delta_{RWU}$) and xylem water at the stem base ($\delta_{xyl.R}$) and at 8 m above the ground ($\delta_{xyl.H}$). **(c)** depicts the FPLDs used to transform the modelled $\delta_{RWU}$ to $\delta_{xyl}$ as depicted in **(a)** & **(b)**. The '*' operator stands for convolution.





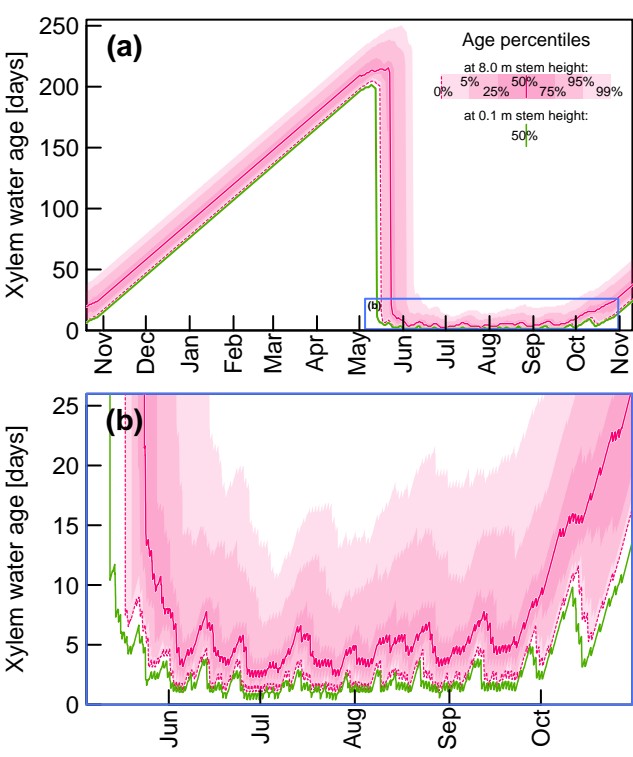

**Figure 8.** Selected quantiles of modelled xylem water ages at 0.1 m stem height (green) and 8 m stem height (pink) based on FPLDs $h_{F1}$ and $h_{F1F2}$, respectively, and time variable sap flow velocities. **(b)** shows the subset of **(a)** which is marked by the blue rectangle.



**Figure A1.** Aggregated CRDS raw data (stable water isotopes and moisture) of all WIP measurements. Solid points indicate median values of daily measurements, while actual values are indicated translucently. Above the plots several incidents are indicated and numbered with letters. In case no symbols are associated to the incident, all probes are affected, otherwise just the ones indicated: A) Initial phase with some unreliable data; B) Intrusion of water (through probe T1R) into the system during a high intensity rainfall event; b) aftermath of B, probe T1R still contains liquid water; c) and C) Effects of strong winds and frail setup lead to loosened electric connections, deactivating the valves of three probes; D) Interruption of dry air supply, no dilution and no through-flow; E) Tubing of probe T2R was severed by rodents, from then on the system was measuring the atmospheric air on this valve slot.





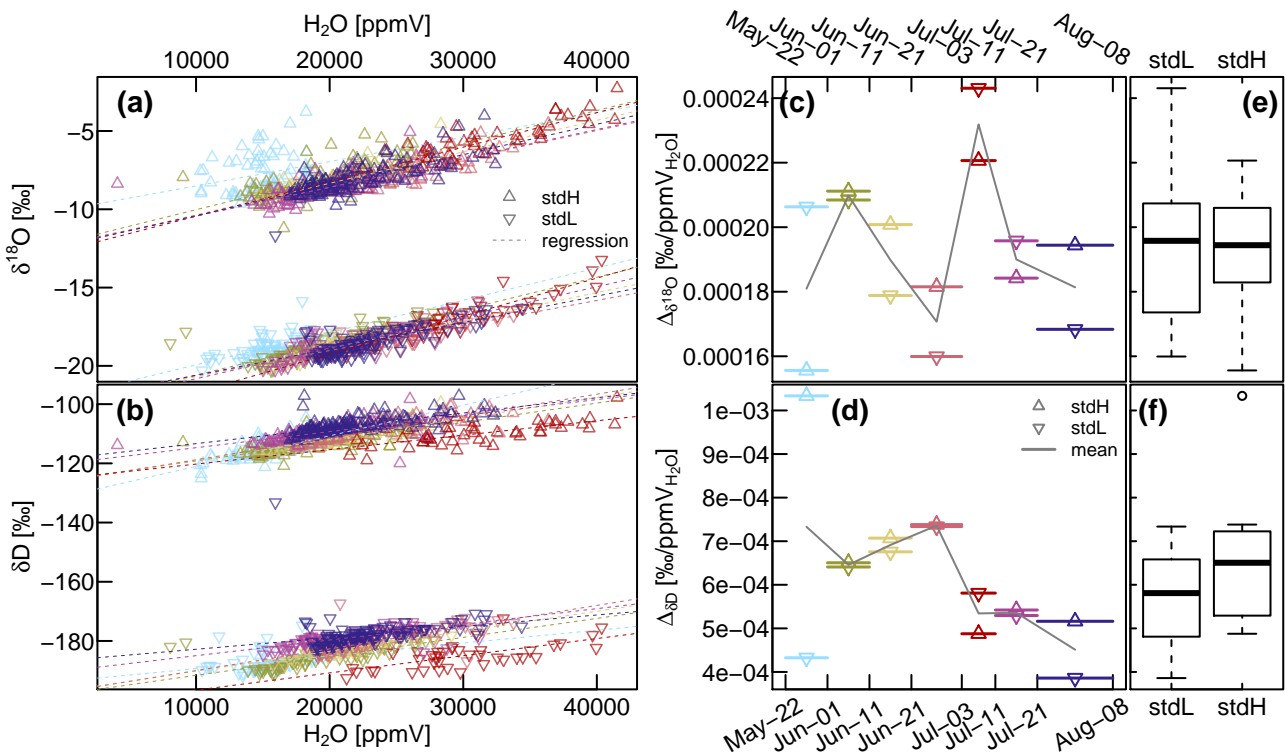

**Figure A2. (a&b)** Relationship between sample gas volumetric moisture content ($H_2O$ [ppmV]) and the measured isotope values assessed with the heavy standard (stdH) and the light standard (stdL). The seven colors represent specific subperiods, as used in **(c)** and **(d)**, which show the variability of the slopes of the regression lines over time. The boxplots shown in subfigures **(e)** and **(f)** summarize this information. The blue period at the end of May indicates some starting problems and the red period at the start of July demonstrates the effect of malfunctioning dilution.



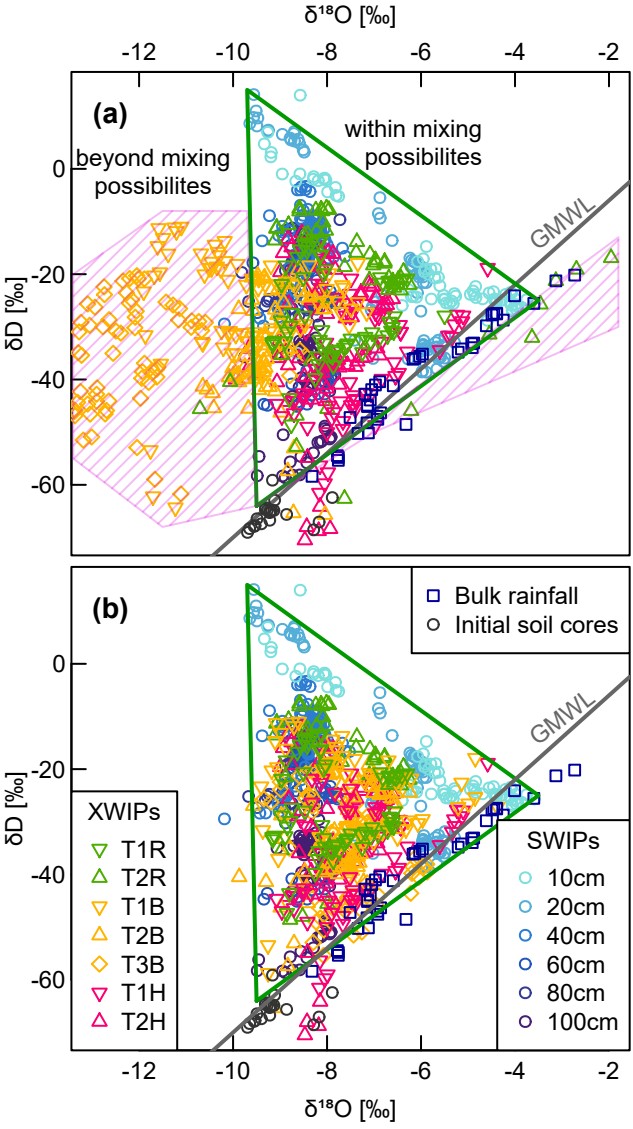

**Figure A3. (a)** Dual isotope plot of the in-situ (XWIPs in tree xylem and SWIPs the soil) measurement data after volumetric moisture correction and normalization to the liquid phase combined with bulk rainfall samples and soil core data measured before the irrigation. The pink shaded area marks the area that contains implausible isotopic xylem signatures, i.e. impossible to obtain by mixtures of the signatures measured within the soil water). Subfigure **(b)** shows the same data after manual removal of some data points and adding one or multiple offsets to the $\delta^{18}O$ time series of T1B. T2B and T3B.

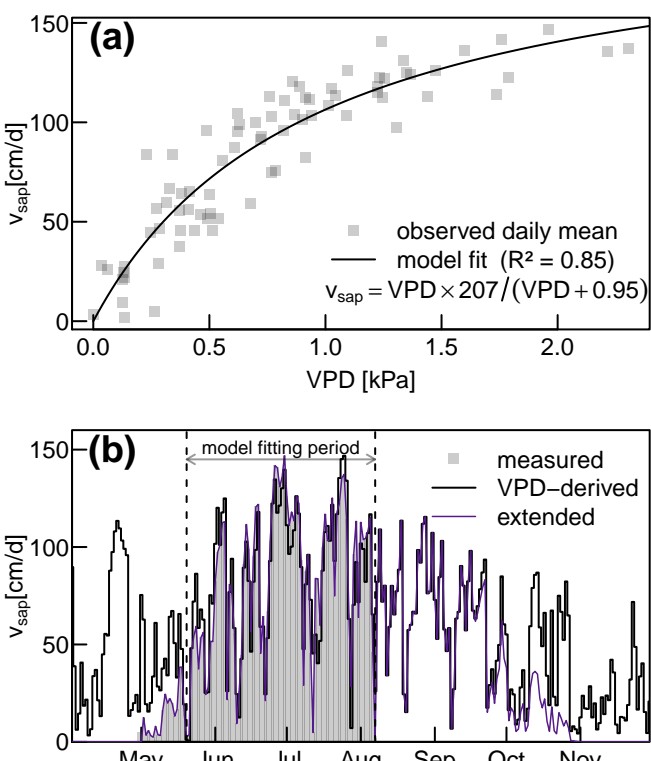

**Figure A4. (a)** Relationship between daily mean values of vapor pressure deficit (VPD) and sap flow velocity ($v_{sap}$). **(b)** Time series of measured and VPD-derived daily mean $v_{sap}$. The model fitting period excludes the time before completion of leaf flush. The purple line shows the extended data as used for the xylem water age computations (values before April and after November are also 0)

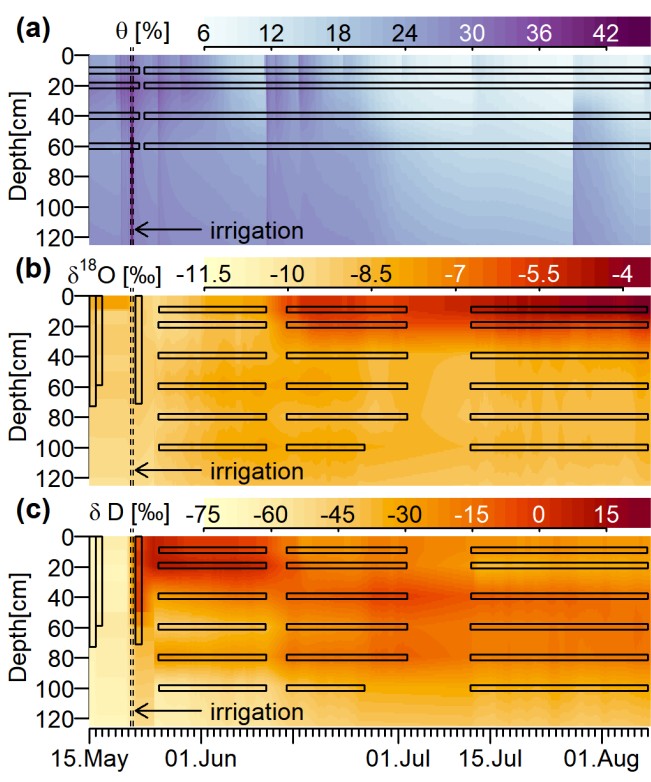

**Figure A5.** Input data for further RWU computations: spatially and temporally interpolated time series of **(a)** volumetric soil moisture, **(b)** Soil $\delta^{18}O$ and **(c)** Soil $\delta D$. Vertical bars represent actual observations obtained by soil core measurements, horizontal bars represent observations measured with probes.



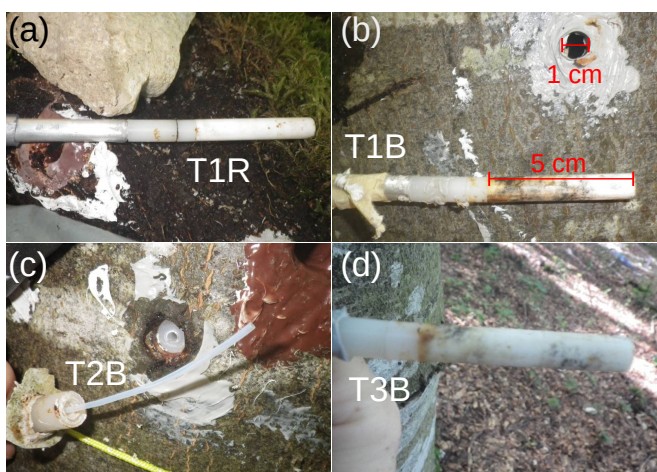

**Figure A6.** Freshly removed XWIPs 12 weeks after installation. The heads of XWIPs T1B **(b)** and T3B **(d)** were covered with biofilms, while the head of T1R**(a)** was as good as new. During removal, the probe head of T2B **(c)** broke off. Callus formation around the drill hole already started.





*Author contributions.* SS and MW designed the experiment. SS conducted the field work and data analysis and wrote the first draft. MW contributed to writing the final manuscript.

*Competing interests.* The authors declare that they have no conflict of interest.

690 *Acknowledgements.* Parts of this work were funded by the DFG research program SPP 1685. Preparation and realization of the irrigation as well as parts of the probe installation were done by Britta Kattenstroth Jonas Schwarz and Michael Rinderer.





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
