# Peer review of "Temporal dynamics of tree xylem water isotopes: In-situ monitoring and modelling"

_Biogeosciences, 2021_

## Referee Comment (RC2)

[referee-annotated manuscript omitted]

---

## Author Comment (AC1)

**RESPONSE TO THE REVIEWER COMMENT OF JOHN MARSHALL REGARDING THE MANUSCRIPT "TEMPORAL DYNAMICS OF TREE XYLEM WATER ISOTOPES: IN-SITU MONITORING AND MODELLING"**

We would like to thank John Marshal for the time he has taken to read our manuscript and his helpful comments to improve it. In the following section we are going to repeat the points brought up (in grey italic letters) and subsequently respond to them:

*The model describes only one pathway from the root to a point in the stem. But the probe is 5 cm long, so there are a tremendous range of fluxes along that radial distance. Generally the flow rates decline whenever the cells become smaller in diameter (latewood of each tree ring) and as the wood ages, i.e., as one moves inward radially. I do not ask for a model that describes all of this, but I think it should be mentioned as a source of variation that the model cannot address.*

We do agree that our simple conceptual model is not suited to explicitly account for the mentioned effects. This is reflected by its inability to properly transform $\delta_{RWU}$ into $\delta_{Xyl}$ based on assumptions about the macroscopically determined flow path length distributions (as described by equations 6 – 9 in section 2.1.2). However, the alternative parametric equation (22) yields a remarkably good fit to the observations of $\delta_{Xyl}$. Even though the fitted shape and scale parameters do not have any obvious relation to physically observable properties of the system, we would say that the resulting apparent flow path length distribution implicitly accounts for macroscopic as well as microscopic effects.

*Sapflux measurements are usually adjusted for the radial trends described in point 1 above. This is important here because so much is made of the comparison of rates. By the way, the depth of sapflux probes was not specified. Perhaps this would help explain some of the discrepancy. At least some recognition of this radial decline issue should be given.*

The utilized probes measure at three depths (5, 17.5 and 30 mm) but our measurements did not show any clear decline of sap flux velocities along those three depths. We will add these details to the revised version of the manuscript. However, we do not directly see how this could explain the observed discrepancy between sap flux and tracer velocities – as the actually shallower penetration depth of the sap flux sensors could only be an explanation for the reverse case of sap flux velocities that exceed the tracer velocities.

*In addition, the heat pulse method is one of the sapflux methods that was found to underestimate gravimetric sapflux in Steppe et al. (2010), who also used Fagus, by the way. The underestimate was not five-fold, but the sensor was not the same brand as this one either. In any case, the issue of calibration should be discussed when the methods are compared.*

This is a really interesting literature suggestion, we have not yet been aware of. We will includ it into our discussion. Indeed, Steppe et al. (2010) show a big discrepancy for heat pulse dissipation probes as used in our experiment, especially without wound correction (as in our case), where actual velocities were 3.7 times higher than the probe based estimates.

*I have always wondered about whether the air leaving these probes is saturated. I was reminded of this question around line 313. I presume that the dry air addition is in part an attempt to prevent saturation, and the condensation that might result. Is that true? In any case, if the interior airspace is not saturated, then it does not seem sufficient to use the equilibrium fractionation to calculate liquid water isotope ratio from that of water vapour. If there is a temperature sensor in the probes, then it may be sufficient to calculate the saturation vapour pressure at the temperature and determine whether the water vapour concentration in the air coming out of the probe is at that concentration. If not, I wonder if a more complex model, accounting for kinetic effects isn't needed to infer the liquid values? I am not convinced by the text on lines 313-315.*

Due to the comparably big surface area of the porous probe head and the small flow rates, the air within the probe head itself is indeed saturated as long as the probe is embedded into a "moist" medium. Volkmann and Weiler (2014) have tested this during summer time in the same region.

The dilution of the sample happens in the mixing chamber (B), not in the probe head (C), and consequently has no effect on fractionation. We revised Fig. 3 to highlight the physical separation of mixing chamber and probe head (see Fig. R1) and will make sure to highlight this distinctive feature of the probe design in the description in Sec. 2.2.3.

[Figure]

**Fig R1: Left side: original version of the proe sketch. Right Side: revised version of the probe sketch with an improved depiction of the probe's specific components.**

**Table 1: Three cases of flow rate partitioning and the resulting effects on the humidity within the mixing chamber (where the sample is taken) and the probe head (where the equilibration happens). The first row represents a flush period, the second row represents a measurement and the third row has been experimentally verified but has no practical use case.**

| Flow rates [mL/min] | | | Humidity within probe compartment | |
|---|---|---|---|---|
| Sample | Dilution | Through-Flow | Mixing chamber | Probe head |
| -35 | 35 | 0 | dry | saturated |
| -35 | 15 | 20 | unsaturated | saturated |
| -35 | 0 | 35 | saturated | saturated |

Since we can assume equilibrium fractionation within our probe heads (see Table 1) and apply the same dilution rate to all our tree-, soil- and standard probes we are confident, that the $H_2O$ concentrations measured by the CRDS can be used to calibrate the field measurements. As shown in Fig. A2, we are getting a fairly consistent relationship between the measured water vapor concentrations and the measured isotopic signatures of our standards. The biggest outliers are found right at the beginning of the experiment (light blue triangles), where we had some initial moisture problems within the tubing of the whole setup and for $^{18}O$ between Jul-03 and Jul-11 (red triangles), where we had problems with the gas supply.

We acknowledge, that our calibration procedure has some space for improvements (see manuscript lines 537 - 542), but for the time being, we would abstain from including a more

complex calibration methodology which would certainly require precise and representative temperature measurements for each single probe head at different depths. This would most certainly be needed to make full use of the sub-daily measurement frequency, but regarding the points treated within this study, a more sophisticated calibration procedure would very likely end up with a rather unfavorable cost-benefit ratio.

*Line 92: these were not logs, but trees cut from their root systems. They drew water up through the stems under tension derived from transpiration, as trees normally do.*

We will correct that in the revised version of the manuscript.

*Line 113: I don't think Jarvis did this with isotopes, did he? Clarify. Also, I think you have a "source strength," not a sink strength.*

We will rewrite Sec 2.1.1 in order to clarify that the computation of $\delta_{RWU}$ is not a part of the Jarvis model and replace "sink strengths" with "source strengths".

*Line 128: a soil layer's* – will be corrected

*Line 234: built* – will be corrected

*Line 386: any idea why a biofilm would change the equilibration?*

We could think of a potential clogging of the membrane head's pores, which might affect the equilibration between the air within and the water outside of the probe head. Additionally, the biofilms might emit volatile organic compounds, which have been shown to influence CRDS-isotope measurements (West et al. 2010, Chang et al. 2016). Unfortunately, the CRDS that we used during that experiment was not logging the respective indicator parameters that could have helped to identify such organic influences on the isotope measurements.

*Line 430: delete "of"* – will be corrected

*Lines 499 to 500: No, the model described in Marshall et al. determined that the water equilibrated within a couple of mm as it passed through the borehole. The issue, if there is one, is the opposite: the borehole vapour represents a thin layer of the sapwood near the outer edge, rather than the whole thing. In that sense, the probes described here may be a better integration of a greater depth. Suggest you say that.*

You may have computed that the passage through 5 mm of borehole is enough to reach an equilibrium with the xylem water, but this does not mean that there is no further exchange with xylem water further along the borehole. According to Marshall et al. (2020), the borehole equilibration technique so far has only been applied to stems with diameters between 8 and 12 cm. At such small diameters, the ratio of conducting to non-conducting parts of the xylem can be expected to be quite different compared to more mature trees. Therefore, we would leave our statement as it is.

*Line 512-518: it seems fair to add that the biofilms may have been favoured by the closed system. There was no such problem with the open boreholes, at least not that we knew about.*

We will include this thought into our discussion (Sec 4.1).

*Lines 569-574: this needs to be reworded. The last sentence is very important, but it means rather little as written. I think what you want to say is that the scaling error allows you to detect trends, but the values are not accurate. Then again, as noted above, I wouldn't place too much faith in the sapflux sensors, so I don't think you know whether they are accurate or not.*

We will revise the whole section of the discussion and hope that it will gain in clarity.

*Line 617-619: I disagree. If the labelled irrigation event were big enough to allow the xylem to come to steady-state for a longish time, then it should be possible to match dRWU against dxyl if there were no other problems with the method. The fact that you couldn't do it doesn't mean it's impossible.*

In the next paragraph (line 620) we already mentioned that under the condition of little temporal variability of the RWU composition $\delta_{RWU}$ and $\delta_{Xyl}$, are similar enough to be treated as equal. And actually this condition seems to be fulfilled most of the time: Fig. 6 shows that there are only three periods (labeled A, B, and C) where $\delta_{Xyl}$ seems to deviate from $\delta_{RWU}$ due to temporal dynamics. All of these periods follow sudden changes of soil water signatures or soil water availability.

We rewrote that part of the conclusion and hope that now it is clear, that we do not generally deny a relation between $\delta_{RWU}$ and $\delta_{Xyl}$, but we want to raise awareness for the possible discrepancy between the two signatures.

References:

West, A.G., Goldsmith, G.R., Brooks, P.D., Dawson, T.E.: Discrepancies between isotope ratio infrared spectroscopy and isotope ratio mass spectrometry for the stable isotope analysis of plant and soil waters. Rapid Commun Mass Spectrom. 2010 Jul 30;24(14):1948–54.

Chang, E., Wolf, A., Gerlein-Safdi, C., Caylor, K.K.: Improved removal of volatile organic compounds for laser-based spectroscopy of water isotopes. Rapid Commun Mass Spectrom. 2016 Mar 30;30(6):784–90.

Volkmann, T. H. M. and Weiler, M.: Continual in situ monitoring of pore water stable isotopes in the subsurface, Hydrology and Earth System Sciences, 18, 1819–1833, https://doi.org/10.5194/hess-18-1819-2014, 2014.

Marshall, J. D., Cuntz, M., Beyer, M., Dubbert, M., and Kuehnhammer, K.: Borehole Equilibration: Testing a New Method to Monitor the Isotopic Composition of Tree Xylem Water in situ, Frontiers in Plant Science, 11, 1–14, https://doi.org/10.3389/fpls.2020.00358, 2020.

Steppe, K., De Pauw, D. J., Doody, T. M., and Teskey, R. O.: A comparison of sap flux density using thermal dis-sipation, heat pulse velocity and heat field deformation methods, Agricultural and Forest Meteorology, 150, 1046–1056,https://doi.org/10.1016/j.agrformet.2010.04.004, http://dx.doi.org/10.1016/j.agrformet.2010.04.004

---

## Author Comment (AC2)

**RESPONSE TO THE REVIEWER COMMENT OF ANONYMOUS REFEREE #2 REGARDING THE MANUSCRIPT "TEMPORAL DYNAMICS OF TREE XYLEM WATER ISOTOPES: IN-SITU MONITORING AND MODELLING"**

We would like to thank the anonymous referee for the time they have taken to read our manuscript and their helpful comments to improve it. The referee has left their comments in a digital copy of the manuscript. The smaller annotations regarding orthography will be implemented in the revised version of the manuscript. We will try to summarize the points made in the additional comments (in grey italic letters) and subsequently respond to them.

*Lines 100 – 102: The referee calls for a science driven motivation of the study.*

We will reformulate the aim of our study into: "*The aim of this study was to investigate the temporal dynamics of xylem water isotopes of mature trees within a field experiment. Making use of the unique possibilities arising from the novel in-situ measurement approach, we wanted to test how far the commonly made assumption of equivalence between the isotopic signatures of RWU and xylem water is true for the case of mature trees.*"

*Line 103: It is not clear to the referee how the hypothesis "Xylem water isotopic signatures are equivalent to the isotopic signature of root water uptake" should be tested as xylem water is the only direct gateway to root water uptake. The referee asks with which other method or approach to determining RWU the xylem method should be compared.*

By reformulating our study aim, we will drop this specific hypothesis.

*Line 138: The referee asks for a derivation or reference to Eq. 6*

Will be answered together with the following comment…

*Lines 141-142: The referee has plotted the Eq. 6 and doubts the statement that a λ value of 1 would lead to a linear decrease of root densities with distance.*

We have to admit, that there is no proper way to derive Eq. 6 in the manuscript. We came up with that equation by iteratively trying out different equations until we fund one which exposed the desired behavior, i.e. an equation that can produce a concave, linear or convex decrease of predicted densities with increasing distance. Apparently, the equation that we actually used differed slightly from the formula depicted in our first manuscript. After the very observant critical referee comment, we revised our choice of the equation and ended up with the following one (behaving similar to the original one, but with a clearer origin):

$$g_r(\mathrm{r}) = 1 - \frac{B_i\left(\frac{r}{r_{max}}, \lambda, 1\right)}{B(\lambda, 1)}$$

This equation is based on the cumulative density function of the beta distribution. $B$ is the beta function while $B_i$ is the incomplete beta function. In this equation, the distance decay parameter λ behaves inversely to what we described in the initial manuscript: values smaller than one indicate a faster than linear density decrease, values bigger than one a slower than linear decrease and λ=1 leads to a linear decrease.

Fig.R1a depicts the influence of different values for the distance decay parameter λ and demonstrates the symmetric behavior of the chosen equation, where 1/ λ produces a mirrored version of the distribution that follows from λ.

[Figure]

**Figure R1: (a) Exemplary depiction of the horizontal fine root density distribution $g_r(r)$ for an $r_{max}$ value of 4 m and a range of different λ values. (b) FPLDs resulting from the same $g_r(r)$ functions as depicted in subfigure (a) and a $z_r$ value of 1 m.**

*Line 157: The referee wants to know, how we accounted for diffusion and or dispersion of the isotopic signal in the xylem especially during periods of low or no sap flow.*

We did not explicitly account for diffusion and dispersion.

*Lines 236-239: The referee asks for more information regarding the materials used to build the probes.*

We will include more information in the description of the probe design and add the utilized materials.

*Line 256: The referee asks for a reference to the Arduino microcontroller platform*

We will add a reference to an introductory article on the topic.

*Line 291: The referee wants to know why the sap flow measurements have been discontinued in early August*

The data logger for the sap flow probes was needed more urgently for another project.

*Line 300: The referee wants to know how comparable the four soil moisture profiles were.*

As can be expected from a skeleton rich loamy-clayey soil, there were some differences between the single profiles. We will add the information that the used soil moisture data was averaged across the four depth profiles.

*Line 337: The referee asks about an estimate of the validity of our assumption of a constant boundary condition for the isotopic signatures in a depth of 2 m*

As already mentioned, we did not have any measurements below the specified depths and our boundary conditions are just assumptions that needed to be made one way or another in order to interpolate our measurement data. Regarding the shallow soil and rooting depths (our optimized RWU model suggest that 95% of all fine roots are found above a depth of 90 cm), which were largely covered by our actual observations, we do not think that the chosen hypothetical values at 2-m depth were overly critical for the final outcome of the study.

*Line 534: The referee asks whether we tested our assumption of water vapor saturation within the probe head and compared measurements to theoretical values.*

Volkmann and Weiler (2014) have tested that for this kind of probe in soil and Marshall et al. (2020) have computed a very fast saturation of added dry air within tree xylem. We have supported our assumption with references.

*Lines 535 to 537: The referee wants to know which safety measure we had to avoid sampling from an unsaturated probe head.*

We did not have any safety measures to avoid sampling from an unsaturated probe head. But Gaj et al. (2016) have used similar soil probes within much dryer soils in the Namib Desert and their comparisons to destructively obtained samples did not indicate any problems that might be caused by sampling from an unsaturated probe head.

In a yet unpublished experiment, where a similar setup with soil probes placed very close to the soil surface (at 2.5, 5.0, 7.5 and 10 cm depth) ran from April to November 2020, all measurements plotted very close to the meteoric water line. This means that even in those parts of the soil that can be expected to become rather dry we did not observe any obviously conspicuous behavior of the probes at any time throughout a whole summer.

*Lines 540 – 541: The referee deems it inevitable to devise a calibration procedure that considers temperatures at each WIP head to make full use of the sub-daily temperature artefacts left in our current calibration procedure.*

We would argue that this kind of enhanced calibration procedure is definitely desirable, but as our measurements show, this kind of measurement setup also may provide valuable insights with a basic calibration procedure, especially in connection with isotopic labeling, but also within the range of natural abundances (i.e. $^{18}O$ during our experiment).

*Line 548: The referee asks how the fact that the root distance decay parameter λ could not be identified affected our model results.*

As can be seen in Fig R1b, the effective FPLDs for small values of λ do not differ much. A λ value of 1/1000 produces more or less the same FPLD as a λ value of 1/20. Even the FPLD resulting from a λ value of 1/5 is still very similar. Actually, ever smaller – in the original manuscript with the old version of the equation for $g_r(r)$ bigger – values for λ lead to slightly better fits. However, at a certain point towards extreme values of λ rounding errors do occur and the quality of the fits deteriorates.

The idea behind the whole approach was to constrain the FPLD by a reasonable assumption on a (potentially measurable) horizontal distribution of fine roots. Eventually, we came to the conclusion, that the tracer inferred FPLD underlies influences (such as dispersion) which cannot be traced back to macroscopically observable characteristics of the root xylem.

*Lines 564-565: The referee asks how diffusion/dispersion have been accounted for.*

We did not explicitly account for diffusion and dispersion, but as mentioned in Lines 581 – 586, the fitted parametric distributions based on Eq. 22 ($h_{F1}$ and $h_{F2}$ in Fig 7c) most certainly are implicitly accounting for dispersion effects.

*Lines 567-568 and 572-573: The referee asks whether diffusion/dispersion could be used to explain the observed discrepancies between sap flux measurements and observed tracer velocities.*

First of all, we could attribute a good part of the observed discrepancy to our lacking wounding correction of $v_{sap}$, but even with such a correction, as John Marshal pointed out in his comment, Steppe et al. (2010) have shown that sap flux sensors based on the heat pulse dissipation method tend to underestimate actual velocities.

Diffusion might become more important during periods of very low or even stagnant sap flow, but the mean sap flow velocities during our experiment probably were too high for unaccounted diffusion to cause observable discrepancies.

References:

Volkmann, T. H. M. and Weiler, M.: Continual in situ monitoring of pore water stable isotopes in the subsurface, Hydrology and Earth System Sciences, 18, 1819–1833, https://doi.org/10.5194/hess-18-1819-2014, 2014.

Marshall, J. D., Cuntz, M., Beyer, M., Dubbert, M., and Kuehnhammer, K.: Borehole Equilibration: Testing a New Method to Monitor the Isotopic Composition of Tree Xylem Water in situ, Frontiers in Plant Science, 11, 1–14, https://doi.org/10.3389/fpls.2020.00358, 2020.

Steppe, K., De Pauw, D. J., Doody, T. M., and Teskey, R. O.: A comparison of sap flux density using thermal dis-sipation, heat pulse velocity and heat field deformation methods, Agricultural and Forest Meteorology, 150, 1046–1056,https://doi.org/10.1016/j.agrformet.2010.04.004, http://dx.doi.org/10.1016/j.agrformet.2010.04.004, 2010.

Ga,j M., Beyer, M., Koeniger, P., Wanke, H., Hamutoko, J., Himmelsbach, T.: In situ unsaturated zone water stable isotope $^2$H and $^{18}$O measurements in semi-arid environments: a soil water balance. Hydrol Earth Syst Sci. 2016 Feb 17;20(2):715–31.

---

## Author Response (AR2)

**RESPONSE TO THE REVIEWER COMMENT OF NICOLAS BRÜGGEMANN REGARDING THE MANUSCRIPT "TEMPORAL DYNAMICS OF TREE XYLEM WATER ISOTOPES: IN-SITU MONITORING AND MODELLING"**

We would like to thank Nicolas Brüggemann for the time he has taken to read our manuscript and his helpful comments to improve it. The technical corrections have been incorporated in the current revision of the paper.

Regarding the criticized lack of discussion of dispersion and diffusion effects:

It lies in the nature of the convolution approach, that dispersion and diffusion cannot explicitly be represented, since the convolution itself is not process based at all. However, by fitting a suited parametric distribution, many effects resulting from physical processes, can roughly be reflected by a convolution. Consequently, we do not think that an explicit incorporation of dispersion and diffusion into our computations lies within the scope of this study.

We have extended section **4.4 Interpretation of FPLDs** to clarify that dispersion is implicitly accounted for within our modelling approach and that a more explicit representation of it would require a whole different type of model.